# Spherical Watermark: Encryption-Free, Lossless Watermarking for Diffusion Models

**Xiaoxiao Hu[1], Jiaqi Jin[1], Sheng Li[1], Wanli Peng[2], Xinpeng Zhang[1], Zhenxing Qian[1,\*]**
[1]Fudan University, [2]China Agricultural University
`{xxhu23,jqjin24}@m.fudan.edu.cn`
`{lisheng,zhangxinpeng,zxqian}@fudan.edu.cn`
`wlpeng@cau.edu.cn`

## Abstract

Diffusion models have revolutionized image synthesis but raise concerns around content provenance and authenticity. Digital watermarking offers a means of tracing generated media, yet traditional schemes often introduce distributional shifts and degrade visual quality. Recent lossless methods embed watermark bits directly into the latent Gaussian prior without modifying model weights, but still require per-image key storage or heavy cryptographic overhead. In this paper, we introduce **Spherical Watermark**, an encryption-free and lossless watermarking framework that integrates seamlessly with diffusion architectures. First, our binary embedding module mixes repeated watermark bits with random padding to form a high-entropy code. Second, the spherical mapping module projects this code onto the unit sphere, applies an orthogonal rotation, and scales by a chi-square-distributed radius to recover exact multivariate Gaussian noise. We theoretically prove that the watermarked noise distribution preserves the target prior up to third-order moments, and empirically demonstrate that it is statistically indistinguishable from a standard multivariate normal distribution. Adopting Stable Diffusion, extensive experiments confirm that Spherical Watermark consistently preserves high visual fidelity while simultaneously improving traceability, computational efficiency, and robustness under attacks, thereby outperforming both lossy and lossless approaches.

## 1 Introduction

Diffusion models have demonstrated transformative potential in creative applications (Rombach et al., 2022; Sahoo et al., 2024), but also raise concerns over authenticity and ownership (Craver et al., 1997; Grinbaum & Adomaitis, 2022). Malicious users can exploit them to fabricate images and spread disinformation, eroding public trust and creating legal and ethical challenges. As governments and platforms face mounting pressure to address harmful content (Biden, 2023; Wiggers, 2023), reliable provenance mechanisms are urgently needed to trace and identify malicious actors.

Image watermarking offers a promising direction by embedding imperceptible identifiers into images. However, traditional schemes alter the data distribution and degrade visual fidelity, whether operating in the spatial (Li et al., 2009; Bender et al., 1995) or frequency (Al-Haj, 2007; Navas et al., 2008) domain. Additionally, some approaches inject watermarks by training or fine-tuning generative models. For example, Fernandez et al. (Fernandez et al., 2023) fine-tune the Stable Diffusion (Rombach et al., 2022) decoder to bake in a hidden mark. To avoid costly retraining and improve flexibility, Wen et al. (Wen et al., 2023) embed ring-patterns in the frequency domains of the latent space. Although robust to lossy transmission, these methods introduce perceptual artifacts and reduced fidelity.

Recently, the concepts of lossless or undetectable watermarking have been proposed. These methods seek to establish an invertible mapping from watermark bits to standard Gaussian noise, embedding watermarks without any modifications to the pretrained generative model. For example, Yang et al. (Yang et al., 2024) introduce Gaussian Shading which uses repeated watermarks and stream cipher for sampling but demands a unique key and nonce per image, *incurring substantial storage*

---

\*Corresponding author.

*and management overhead.* Gunn et al. (Gunn et al., 2025) later replace the stream cipher with fixed-key pseudorandom error-correcting codes (Christ & Gunn, 2024). Nonetheless, the heavyweight cryptographic constructs also introduce drawbacks: *they incur nontrivial computational and decoding latency, demand careful parameter tuning to balance code rate and error-correction capability, and fail under strong attacks that exceed the code's designed distortion bounds.*

In this paper, we propose Spherical Watermark, a simple yet effective lossless scheme that is encryption-free and robust against common attacks. Our method integrates seamlessly with pretrained diffusion models via three modules: binary embedding, spherical mapping, and diffusion integration. The binary embedding module mixes watermark bits with random paddings to produce a 3-wise independent bitstream. The spherical mapping module then projects this bitstream onto the unit sphere, applies an orthogonal rotation, and scales it by a chi-square-distributed radius. We theoretically analyze each intermediate distribution and prove that the final noise is statistically indistinguishable from standard Gaussian noise. In addition, our encryption-free design eliminates the need for per-image key storage. The diffusion integration module then feeds the watermarked noise into Stable Diffusion (Rombach et al., 2022) to produce watermarked images. Experiments show that our scheme preserves fidelity and surpasses lossy methods. Compared to lossless approaches (Gunn et al., 2025), our method offers stronger traceability, reduced complexity, and enhanced reliability.

In summary, our key contributions are three-folded: 1) We propose a novel lossless watermarking framework, which seamlessly integrates with diffusion-based architectures. Our method guarantees robust watermark extraction while preserving the original generation fidelity. 2) We introduce a simple yet effective mapping strategy that transforms binary watermarks into Gaussian noise inputs. We provide both theoretical analysis and empirical evidence that the watermarked noise distribution is statistically indistinguishable from a standard multivariate normal distribution. 3) Compared to existing lossless watermarking schemes, our encryption-free approach omits key storage overhead, enabling an excellent trade-off between undetectability and watermark robustness.

## 2 RELATED WORKS

Digital image watermarking has been extensively studied to safeguard intellectual property. Traditional watermarking methods can be applied directly to diffusion outputs, whether operating in the spatial domain (Li et al., 2009; Bender et al., 1995), the frequency domain (Navas et al., 2008; Liu et al., 2017; Kashyap & Sinha, 2012), or via neural-network embedding (Zhang et al., 2019; Zhu et al., 2018; Tancik et al., 2020). In addition, several works embed watermarks by fine-tuning diffusion models (Fernandez et al., 2023; Xiong et al., 2023; Kim et al., 2024; Wang et al., 2025). For example, SleeperMark (Wang et al., 2025) introduces a trigger mechanism to decouple watermark information from semantic content, keeping the watermark extractable after model fine-tuning. More recently, latent-based watermarking has gained attention. Wen et al. propose the Tree-Ring (Wen et al., 2023) watermarking scheme, which embeds ring-shaped patterns into frequency domains of the latent space to enable detection. Subsequent works such as RingID (Ci et al., 2024), SEAL (Arabi et al., 2025b), and WIND (Arabi et al., 2025a) design alternative patterns. Beyond pattern-based designs, Wei et al. (Wei et al., 2024) provide a unified analytical framework for diffusion watermarking and instantiate several distribution-preserving schemes, including truncated Gaussian sampling and Gaussian ring watermarking. However, these methods are limited to merely verifying the presence of watermark, not supporting large-scale provenance.

To overcome this limitation, Yang et al. (Yang et al., 2024) introduce Gaussian Shading, a provably lossless watermarking method that employs repetition codes and a stream cipher to sample from the standard Gaussian distribution. However, the reliance on a distinct cipher key and nonce for each generated image imposes a huge key-management overhead that is impractical in the real world. Gunn et al. (Gunn et al., 2025) advocate replacing the stream cipher with the pseudorandom error-correcting codes (PRC) (Christ & Gunn, 2024), which allow the generation of distinct pseudorandom sequences from a fixed secret key. PRC's extensive cryptographic operations also introduce several challenges. Encoding and belief-propagation decoding (Pearl, 2014) incur substantial computation and latency. Finding a trade-off between code rate and error-correction strength requires careful tuning. Moreover, under aggressive post-processing or shifts in the data distribution, the scheme can hit an irreducible error floor and fail to recover the watermark. In this paper, we introduce Spherical Watermark, a framework that eliminates per-image key management, ensures lossless watermark embedding, and demonstrates superior robustness with high computational efficiency.

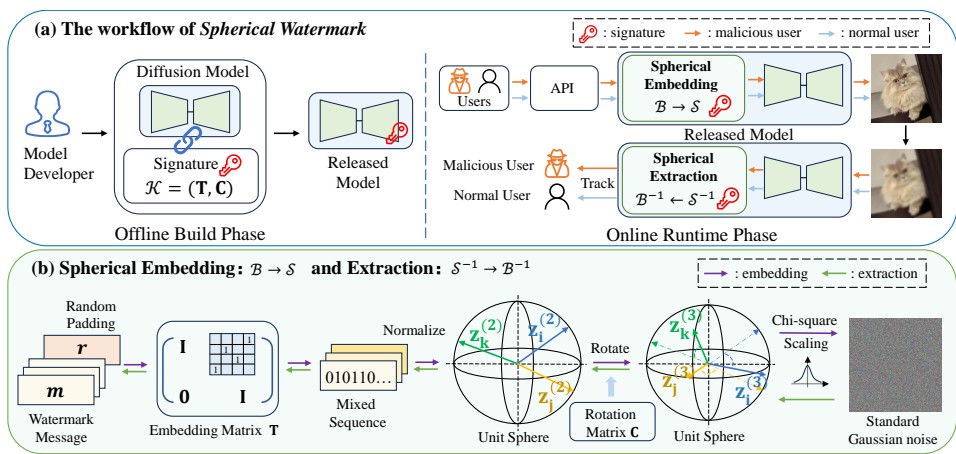

Figure 1: The overall pipeline of our framework.

# 3 METHOD

As illustrated in Figure 1(a), our method constructs a tracing mechanism from the model developer's perspective. In the offline build phase, the model developer generates a fixed "Signature", a set of invertible transforms that encode distinct binary watermarks into the diffusion model's Gaussian noise input. During the online runtime phase, API-driven image request automatically applies the same signature to embed a user-related watermark into the latent code before it is passed through the diffusion model, ensuring that synthesized images carry traceable provenance. Finally, the developer inverts generated images to extract watermarks for reliable provenance tracking.

## 3.1 PROBLEM FORMULATION

The secret watermark $\mathbf{m}$ encodes API metadata (e.g., user ID, timestamp). Let $\mathcal{G} : \mathbb{R}^{l_x} \to \mathcal{I}$ denote a fixed, pretrained diffusion generator that maps standard Gaussian noise $\mathbf{z}$ to a generated image $\mathbf{O}$. Since diffusion models admit an approximate inverse mapping, we use $\mathcal{G}^{-1}$ to recover the latent representation from a generated image. Assume the watermark length is $l_m$. Our goal is to design two efficient procedures in the latent space:

$$\text{Embed} : \mathbf{m} \in \{0, 1\}^{l_m} \ \to \ \mathbf{z}_w \in \mathbb{R}^{l_x}, \qquad \text{Extract} : \hat{\mathbf{z}}_w \in \mathbb{R}^{l_x} \ \to \ \hat{\mathbf{m}} \in \{0, 1\}^{l_m}. \tag{1}$$

Specifically, Embed takes $\mathbf{m}$ to produce the watermarked latent $\mathbf{z}_w = \text{Embed}(\mathbf{m})$, and Extract predicts $\hat{\mathbf{m}}$ from the inverted latent $\hat{\mathbf{z}}_w = \mathcal{G}^{-1}(\mathbf{O}_w)$, where $l_x$ denotes the latent dimensionality of $\mathbf{z}_w$ and $\mathbf{O}_w$ is the generated image with tracable watermark. Let $\Pr[\cdot]$ denotes probability, and $\text{negl}(\rho)$ is a function that vanishes faster than any inverse polynomial in the security parameter $\rho$. We require:

**Undetectability (Losslessness).** For any probabilistic polynomial-time adversary $A$,

$$\big|\Pr[A(\mathbf{z}_w) = 1] \ - \ \Pr[A(\mathbf{z}) = 1]\big| \ \leq \ \text{negl}(\rho). \tag{2}$$

In other words, watermarked noise $\mathbf{z}_w$ is computationally indistinguishable from standard Gaussian noise $\mathbf{z}$. Thus, for any polynomial-time adversary $A'$, the generated images remain indistinguishable:

$$\big|\Pr\big[A'(\mathcal{G}(\mathbf{z}_w)) = 1\big] \ - \ \Pr\big[A'(\mathcal{G}(\mathbf{z})) = 1\big]\big| \ \leq \ \text{negl}(\rho). \tag{3}$$

**Traceability (Exact Extraction).** There exists an Extract such that, given watermarked image $\mathbf{O}_w$,

$$\Pr\big[\text{Extract}(\mathcal{G}^{-1}(\mathbf{O}_w)) = \mathbf{m}\big] \ \geq \ 1 - \text{negl}(\rho). \tag{4}$$

That is, the recovered watermark matches the original except with only negligible error in $\rho$.

For watermarking generated samples, losslessness is the central design principle. It preserves visual fidelity and underpins robustness in adversarial settings. We formally justify this in Appendix E and provide empirical evidence in Section 4.2, showing that lossy watermarking can be easily broken by adversarial attacks, whereas lossless watermarking remains unaffected.

## 3.2 METHODOLOGICAL DESIGN

**Watermark Preprocessing.** We represent watermark $\mathbf{m}$ as independent Bernoulli($\frac{1}{2}$) bits. To enhance randomness and error correction, we repeat $\mathbf{m}$ across $N$ blocks and append a padding vector $\mathbf{r} \in \{0,1\}^{l_r}$, drawn i.i.d. from a Bernoulli($\frac{1}{2}$) distribution on each invocation. The resulting vector

$$\mathbf{x} = \begin{bmatrix} \mathbf{m} & \mathbf{m} & \cdots & \mathbf{m} & \mathbf{r} \end{bmatrix}^\top \in \{0,1\}^{l_x}, l_x = N \times l_m + l_r, \tag{5}$$

serves as the sole input to the subsequent transforms.

**Build Phase.** In the build phase, the model developer constructs the *Signature* $\mathcal{K} = \{\mathbf{T}, \mathbf{C}\}$. To reduce the correlation introduced by repeating $\mathbf{m}$, we inject randomness from the padding vector $\mathbf{r}$. Accordingly, the embedding matrix $\mathbf{T} \in \{0,1\}^{l_x \times l_x}$ is designed to mix watermark bits with random paddings while remaining invertible. The rotation matrix $\mathbf{C}$, also invertible, then maps the binary sequence into Gaussian-like noise. $\mathcal{K}$ is kept fixed and secret during runtime to prevent unauthorized removal. The embedding matrix $\mathbf{T}$ is constructed from the identity matrices $\mathbf{I}_{l_{Nm}}$ and $\mathbf{I}_{l_r}$ of sizes $l_{Nm}$ and $l_r$, together with a sparse binary matrix $\mathbf{R} \in \{0,1\}^{l_{Nm} \times l_r}$ generated by Algorithm 1:

$$\mathbf{T} = \begin{bmatrix} \mathbf{I}_{l_{Nm}} & \mathbf{R} \\ \mathbf{0} & \mathbf{I}_{l_r} \end{bmatrix}, l_{Nm} = N \times l_m. \tag{6}$$

The core design lies in $\mathbf{R}$, which injects randomness from the padding vector into the watermark bits. Two parameters govern this construction. The row sparsity $s$ specifies how many random paddings each watermark bit is mixed with: a larger $s$ improves indistinguishability at the cost of amplified error propagation (see Section 4.3). In addition, redundancy is introduced through $N$ repetitions, which enable majority vote decoding. Algorithm 1 ensures that the $N$ copies of each bit are mixed with disjoint subsets of paddings, guaranteeing the independence property proved in Theorem 3.1.

---

**Algorithm 1** Construction of Binary Matrix $\mathbf{R}$

---

**Require:** Positive integers $N, l_m, l_r, s$ such that $l_r \geq N \times s$
**Ensure:** Binary matrix $\mathbf{R} \in \{0,1\}^{l_{Nm} \times l_r}$, Indices Set $P$
1: **Initialize** $\mathbf{R} \leftarrow \mathbf{0}^{N \times l_m \times l_r}$
2: $P \leftarrow \emptyset$
3: **for** $j = 1$ to $l_m$ **do**
4: $\quad \pi \leftarrow \text{RandomPermutation}([1, 2, \ldots, l_r])$
5: $\quad TMP \leftarrow \pi[1 : N \times s]$
6: $\quad$ **for** $i = 1$ to $N$ **do**
7: $\quad\quad G \leftarrow TMP[(i-1) \times s + 1 : i \times s]$
8: $\quad\quad \mathbf{R}[i, j, G] \leftarrow 1$
9: $\quad\quad P \leftarrow P \cup \{(i, j, G)\}$
10: $\quad$ **end for**
11: **end for**
12: **Return** Reshape($\mathbf{R}, (l_{Nm}, l_r)$), $P$

---

By design, $\mathbf{T}$ is bijective over the binary field $\mathbb{F}_2$ and its inverse $\mathbf{T}^{-1}$ follows that $\mathbf{T}^{-1} = \mathbf{T}$. And the rotation matrix $\mathbf{C} \in \mathbb{R}^{l_C \times l_C}$ is orthogonal, so its inverse satisfies $\mathbf{C}^{-1} = \mathbf{C}^T$. We obtain $\mathbf{C}$ by drawing a matrix $l_C \times l_C$ with i.i.d. $\mathcal{N}(0,1)$ and then applying a QR decomposition, retaining the orthogonal factor. $\mathbf{C}$ maps the binary sequence into a continuous noise compatible with the latent input of diffusion models. For notational convenience, we set $l_C = l_x$ in the following descriptions[1].

**Runtime Phase.** Latent-based diffusion models adopt the encoder and decoder of VAE (Kingma & Welling, 2014) to construct bidirectional mappings between the latent and pixel space.

$$E_{\text{VAE}} : \mathcal{I} \to \mathbb{R}^{l_x}, \quad D_{\text{VAE}} : \mathbb{R}^{l_x} \to \mathcal{I}, \tag{7}$$

denote the pretrained VAE encoder and decoder, respectively. Let $\mathbf{z}_T$ be standard Gaussian noise in latent space, and let $\mathbf{z}_0 = E_{\text{VAE}}(\mathbf{O})$ denote the clean latent encoding of an image $\mathbf{O}$. To transform $\mathbf{z}_T$ into $\mathbf{z}_0$, the diffusion model iteratively perform denoising steps over $T$ discrete timesteps:

---

[1]In practice, $l_C$ is chosen as a factor of $l_x$ (e.g. $l_C = \lfloor \sqrt{l_x} \rfloor$) to balance rotational expressiveness with computational and storage efficiency.

$\mathbf{z}_T \rightarrow \mathbf{z}_{T-1} \rightarrow \cdots \rightarrow \mathbf{z}_0$. At each diffusion timestep, the marginal distribution of $\mathbf{z}_t$ is governed by the probability-flow ordinary differential equation (ODE) (Song et al., 2021b):

$$\frac{d\mathbf{z}_t}{dt} = f_t(\mathbf{z}_t) \;-\; \tfrac{1}{2}\, g_t^2\, \nabla_{\mathbf{z}_t} \log p_t(\mathbf{z}_t), \tag{8}$$

where $f_t$ and $g_t$ are drift and diffusion coefficients determined by the pre-defined noising schedule. The score function $\nabla_{\mathbf{z}_t} \log p_t(\mathbf{z}_t)$ is approximated by a neural network $s_\theta(\mathbf{z}_t, t)$. We now describe how watermark embedding and extraction are seamlessly integrated into the Stable Diffusion pipeline.

Our approach decomposes into three reversible modules: Binary Embedding Module $\mathcal{B}$, Spherical Mapping Module $\mathcal{S}$, and Diffusion Integration Module $\mathcal{G}$. As illustrated in Figure 1(b), for watermarked image generation, we first construct the preprocessed input $\mathbf{x}$ by repeating $\mathbf{m}$ and appending random padding $\mathbf{r}$. Then binary embedding module $\mathcal{B}$ performs the matrix multiplication

$$\mathbf{z}^{(1)} \;=\; \mathbf{T}\mathbf{x} \tag{9}$$

in $\mathbb{F}_2$. Next, spherical mapping module $\mathcal{S}$ converts $\mathbf{z}^{(1)} \in \{0,1\}^{l_x}$ into Gaussian noise by

$$\mathbf{v} = 2\mathbf{z}^{(1)} - \mathbf{1}, \mathbf{z}^{(2)} = \frac{\mathbf{v}}{\|\mathbf{v}\|_2}, \mathbf{z}^{(3)} = \mathbf{C}\mathbf{z}^{(2)},$$
$$\text{draw } r \text{ such that } r^2 \sim \chi^2(l_x), \mathbf{z}_w = r\,\mathbf{z}^{(3)}. \tag{10}$$

Here, $\|\cdot\|_2$ denotes the Euclidean norm, and $\chi^2(l_x)$ is the chi-square distribution with $l_x$ degrees of freedom. The diffusion integration module $\mathcal{G}$ then generates the watermarked image. We set the initial noise $\mathbf{z}_T = \mathbf{z}_w$, and by solving Eq. 8 from $t = T$ to $t = 0$, recover the clean latent $\mathbf{z}_0$ from $\mathbf{z}_T$,

$$\mathbf{z}_0 = \text{ODESolve}\big(\mathbf{z}_T;\, s_\theta,\, \text{cond},\, T,\, 0\big). \tag{11}$$

Here, $\text{cond}$ denotes sampling conditions (e.g. text prompts), and ODESolve may be instantiated with different solvers such as DDIM (Song et al., 2021a), DPM-Solver (Lu et al., 2022; 2025), or other ODE integrators. $\mathbf{z}_0$ is then passed through $D_{\text{VAE}}$ to produce the watermarked image $\mathbf{O}_w = D_{\text{VAE}}\big(\mathbf{z}_0\big)$.

For watermark extraction, the developer applies the inverse modules in the order $\mathcal{G}^{-1}, \mathcal{S}^{-1}, \mathcal{B}^{-1}$ on a suspect image $\hat{\mathbf{O}}_w$. Specifically, the developer uses $E_{\text{VAE}}$ to estimate the latent $\hat{\mathbf{z}}_0 = E_{\text{VAE}}(\hat{\mathbf{O}}_w)$, and then solves Eq. 8 from $t = 0$ to $t = T$ to obtain an estimate of the initial noise:

$$\hat{\mathbf{z}}_T = \text{ODESolve}\big(\hat{\mathbf{z}}_0;\, s_\theta,\, \varnothing,\, 0,\, T\big). \tag{12}$$

Here, $\varnothing$ denotes the empty condition (no text prompt). Finally, the developer inverts $\hat{\mathbf{z}}_T$ as,

$$\hat{\mathbf{z}}^{(2)} = \mathbf{C}^{-1}\hat{\mathbf{z}}_T, \hat{\mathbf{z}}^{(1)} = \text{round}\big(\tfrac{\hat{\mathbf{z}}^{(2)}+\mathbf{1}}{2}\big), \hat{\mathbf{x}} = \mathbf{T}^{-1}\hat{\mathbf{z}}^{(1)}, \tag{13}$$

where $\text{round}(\cdot)$ refers to the rounding operation. The first $l_{Nm}$ entries of $\hat{\mathbf{x}}$ correspond to $N$ repeated copies of the watermark message. We therefore apply a majority-vote rule across each group of $N$ bits to obtain the final decoded watermark $\hat{\mathbf{m}}$. To avoid ties, $N$ is chosen to be odd. Our embedding and extraction pipeline guarantees high-precision watermark retrieval for reliable provenance tracking.

### 3.3 THEORETICAL ANALYSIS

In this section, we provide theoretical guarantees that, after the successive mappings $\mathbf{x} \rightarrow \mathbf{z}^{(1)} \rightarrow \mathbf{z}^{(2)} \rightarrow \mathbf{z}^{(3)} \rightarrow \mathbf{z}_w$, the final latent code $\mathbf{z}_w$ is distributed as $\mathcal{N}(\mathbf{0}, \mathbf{I}_{l_x})$ in $\mathbb{R}^{l_x}$. The detailed proofs of all lemmas and theorems stated in this section are provided in the Appendix C.

First, we analyze the distribution of $\mathbf{z}^{(1)}$ in Theorem 3.1. By introducing $\mathbf{r}$ and carefully designing $\mathbf{T}$, we ensure that the resulting high-entropy code $\mathbf{z}^{(1)}$ exhibits strong independence properties.

**Theorem 3.1.** *If $\mathbf{m}$ and $\mathbf{r}$ consist of independent* $\text{Bernoulli}(\tfrac{1}{2})$ *bits, then for $\mathbf{z}^{(1)}$ in Eq. 9, we have* $z_i^{(1)} \sim \text{Bernoulli}(\tfrac{1}{2})$ *for every $i \in \{1, \ldots, l_x\}$, and $\mathbf{z}^{(1)}$ is both 2-wise and 3-wise independent.*

Building on the properties established in Theorem 3.1, we show that $\mathbf{z}^{(2)}$ satisfies the conditions of a spherical 3–design. A spherical $t$-design (Bannai, 1979; Bajnok, 1992) is a finite set of points on the unit sphere that, *up to degree $t$*, exactly matches the averages of all real polynomials with those of the continuous uniform distribution. Consequently, it can be regarded as an *approximate* uniform distribution on the unit sphere. The rigorous mathematical definition of a spherical $t$–design is as,

**Definition 3.1** (Spherical $t$-Design). *A finite set of points $X = \{x_1, \ldots, x_N\} \subset S^{n-1}$ on the unit sphere in $\mathbb{R}^n$ is called a spherical $t$-design if, for every real polynomial $f$ of total degree at most $t$,*

$$\frac{1}{N} \sum_{x \in X} f(x) = \frac{1}{|S^{n-1}|} \int_{S^{n-1}} f(x)\, d\sigma(x),$$

*where $d\sigma$ is the uniform surface measure on $S^{n-1}$, and $|S^{n-1}|$ denotes the total surface area of the unit $(n-1)$-sphere.*

Equivalently, $X$ is a $t$-design if and only if it *matches all moments* of the uniform distribution on the sphere up to degree $t$. Consequently, a spherical $t$-design is indistinguishable from the uniform distribution on $S^{n-1}$ by any statistic of degree $\leq t$, and thus may be viewed as an *approximation* to the uniform spherical distribution. We derive that the set of $\mathbf{z}^{(2)}$ is a spherical 3-design in Theorem 3.2.

**Theorem 3.2.** $\mathbf{z}^{(2)}$ *satisfies that each $z_i^{(2)}$ takes values $\pm \frac{1}{\sqrt{l_x}}$ with $\Pr[z_i = +\frac{1}{\sqrt{l_x}}] = \Pr[z_i = -\frac{1}{\sqrt{l_x}}] = \frac{1}{2}$, $i \in (1, \cdots, l_x)$; $\mathbf{z}^{(2)}$ is 2-wise and 3-wise independent. Then the finite set of $\mathbf{z}^{(2)}$ on the unit sphere $S^{l_x-1}$ is a spherical 3–design.*

Finally, the following two lemmas analyze the distributions of $\mathbf{z}^{(3)}$ and $\mathbf{z}_w$. In Lemma 3.3 we show that the orthogonally rotated vector $\mathbf{z}^{(3)}$ remains uniformly distributed on $S^{l_x-1}$. In Lemma 3.4 we prove that scaling by $r \sim \chi(l_x)$ yields $\mathbf{z}_w = r\, \mathbf{z}^{(3)} \approx \mathcal{N}(\mathbf{0}, \mathbf{I}_{l_x})$ in $\mathbb{R}^{l_x}$. The detailed proofs are given in the Appendix C, and our experiments confirm that the empirical distribution of $\mathbf{z}_w$ is statistically indistinguishable from standard Gaussian distribution in Section 4.2.

**Lemma 3.3.** *Let $\mathbf{z}^{(2)} \in S^{l_x-1}$ be a spherical 3–design. If we apply a fixed orthogonal rotation $\mathbf{z}^{(3)} = \mathbf{C}\, \mathbf{z}^{(2)}$, then $\mathbf{z}^{(3)}$ is also a spherical 3–design. For each coordinate $z_i^{(3)}$, one has $\mathbb{E}[z_i] = 0$ and $\mathbb{E}[z_i^2] = 1/l_x$, and as $l_x \to \infty$, the marginal law of $z_i^{(3)}$ converges to $\mathcal{N}(0, 1/l_x)$.*

**Lemma 3.4.** *Let $\mathbf{n} \sim \mathcal{N}(\mathbf{0}, \mathbf{I}_n)$ be a standard multivariate normal vector in $\mathbb{R}^n$. Then $\mathbf{n}$ admits a polar decomposition of the form*

$$\mathbf{n} = r \cdot \mathbf{u},$$

*where $r^2 \sim \chi^2(n)$, and $\mathbf{u}$ is uniformly distributed on the unit sphere $S^{n-1}$. Furthermore, $r$ and $\mathbf{u}$ are statistically independent. Conversely, if $r^2 \sim \chi^2(n)$, $\mathbf{u}$ is uniformly distributed on $S^{n-1}$, and $r \perp \mathbf{u}$, then the product $\mathbf{n} = r \cdot \mathbf{u}$ follows a standard multivariate normal distribution, i.e., $\mathbf{n} \sim \mathcal{N}(\mathbf{0}, \mathbf{I}_n)$.*

## 4 EXPERIMENT

### 4.1 EXPERIMENTAL SETTINGS

**Implementation Details.** We adopt Stable Diffusion (SD) v1.5[2] and v2.1[3] as backbone generative models. Generated images are $512 \times 512$ color images with latent size $4 \times 64 \times 64$. During the diffusion process, we use a 50-step DPM-Solver++ (Lu et al., 2025) for image generation with a guidance scale of 7.5 and a 50-step DDIM inversion (Song et al., 2021a) with a guidance scale of 1.0. To simulate real-world scenarios, DDIM inversion uses empty prompts. Default settings are $N = 31$, $l_m = 512$, $l_r = 512$, and $s = 1$, giving $l_{Nm} = 15872$ and $l_x = 16384$, which matches the diffusion latent dimensionality. All experiments are conducted in PyTorch on four NVIDIA RTX 4090 GPUs.

**Watermark baselines.** We consider the following baselines: traditional watermarking methods include DwtDct (Al-Haj, 2007), DwtDctSvd (Navas et al., 2008), and RivaGAN (Zhang et al., 2019), all configured to embed 32-bit watermarks. Latent-based baselines include Tree-Ring (Wen et al., 2023), Gaussian Shading (Yang et al., 2024), and PRC Watermark (Gunn et al., 2025). All schemes are evaluated with 512-bit watermarks, except Tree-Ring, which supports detection only. For latent-based methods, we generate five fixed keys (or signatures) and report the mean and standard deviation of each metric over five independent runs. Unless noted otherwise, baselines use their default settings. Note that with fixed keys, Gaussian Shading no longer achieves true losslessness.

**Datasets & Evaluation metrics.** For text prompts, we use two datasets, termed COCO and SDP. Each comprises 1000 text prompts randomly sampled from the MS-COCO val2017 set (Lin et al., 2014)

---

[2] https://huggingface.co/stable-diffusion-v1-5/stable-diffusion-v1-5
[3] https://huggingface.co/stabilityai/stable-diffusion-2-1-base

Table 1: FID value for different watermarking methods. Lower FID indicates higher image quality. $Mean_{Std}$ represents the mean value with 1-sigma error bar.

| Method | COCO | | SDP | |
|---|---|---|---|---|
| | SD v1.5 | SD v2.1 | SD v1.5 | SD v2.1 |
| Original | $48.1256_{1.3744}$ | $46.8146_{1.0617}$ | $49.7041_{0.5425}$ | $46.4060_{0.5231}$ |
| DwtDct | $48.2975_{1.3918}$ | $46.9771_{1.0702}$ | $49.9853_{0.5385}$ | $46.7304_{0.5163}$ |
| DwtDctSvd | $48.7179_{1.4075}$ | $47.4049_{1.0121}$ | $51.0160_{0.6162}$ | $47.5044_{0.6439}$ |
| RivaGan | $48.7956_{1.3952}$ | $47.6124_{1.1012}$ | $51.2773_{0.6320}$ | $47.8298_{0.6748}$ |
| Tree-Ring | $49.3318_{1.5108}$ | $47.8721_{1.1320}$ | $50.6491_{1.0197}$ | $47.3917_{0.7127}$ |
| Gaussian Shading | $50.6968_{1.3200}$ | $49.4379_{1.0326}$ | $51.5221_{0.8773}$ | $48.2539_{0.4859}$ |
| PRC Watermark | $48.1348_{1.3074}$ | $46.7544_{1.0748}$ | $49.5250_{0.7651}$ | $46.4157_{0.3445}$ |
| Ours | $48.1224_{1.5489}$ | $46.8132_{1.0962}$ | $49.3894_{0.7475}$ | $46.4311_{0.3695}$ |

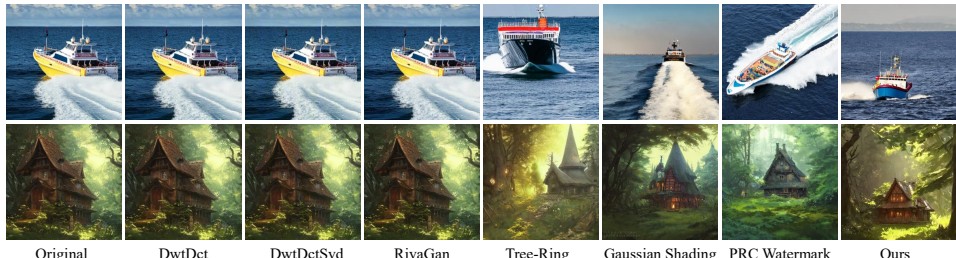

Figure 2: Classification performance over training epochs for distinguishing watermarked from unwatermarked samples. Left Two: Training loss and test accuracy at latent-level. Right Two: Training loss and test accuracy at image-level on SDP dataset with SD v2.1.

| Original | DwtDct | DwtDctSvd | RivaGan | Tree-Ring | Gaussian Shading | PRC Watermark | Ours |

Figure 3: Examples of different watermarking methods. Top: COCO dataset. Bottom: SDP dataset.

and the Stable Diffusion Prompt dataset[4], respectively. To evaluate the performance of our method, we focus on two core criteria: undetectability and tracing accuracy. For undetectability, we first assess any degradation introduced by watermark embedding. To detect subtle distributional shifts, we employ the Fréchet Inception Distance (FID) (Heusel et al., 2017) measured against the unwatermarked output distribution. We also train binary classifiers on both image-level pixels and latent-space inputs to distinguish watermarked from non-watermarked samples, reporting classification accuracy to reveal detectable artifacts introduced by the watermark embedding. Next, we evaluate the reliability of watermark extraction for 100 distinct users under common storage and transmission degradations, including post-processing attacks and adversarial attacks from WEvade (Jiang et al., 2023). Extraction performance is quantified by bit-level accuracy (ACC) and the true positive rate at $1\%$ false positive rate (TPR@1%FPR). For simplicity, we abbreviate TPR@1%FPR as TPR in the sequel.

We report mean and standard deviation over five runs for all metrics. Additional experimental results are provided in Appendix F, including further undetectability experiments and ablation studies.

## 4.2 PERFORMANCE ANALYSIS

**Undetectability.** To assess undetectability, we train classifiers to capture distributional shifts. First, we train a two-layer MLP (Rumelhart et al., 1986) for latent-level classification. According to Figure 2, both Tree-Ring and Gaussian Shading (with fixed keys) are easily detected with accuracies of 100% and 97%, while PRC Watermark and our method remain indistinguishable. Second, we

---

[4]https://huggingface.co/datasets/Gustavosta/Stable-Diffusion-Prompts

Table 2: Comparison results on ACC and TPR. Dataset: COCO dataset. Model: SD v2.1. "Post." refers to "Post-Processing" and "Adv." refers to "Adversarial".

| Method | Metrics | | | | | |
|---|---|---|---|---|---|---|
| | ACC (Clean) | ACC (Post.) | ACC (Adv.) | TPR (Clean) | TPR (Post.) | TPR (Adv.) |
| DwtDct | $90.14_{1.15}$ | $64.75_{1.08}$ | $49.28_{0.00}$ | $92.80_{3.14}$ | $52.23_{3.41}$ | $16.15_{0.02}$ |
| DwtDctSvd | $100.00_{0.00}$ | $93.21_{0.17}$ | $48.95_{0.01}$ | $100.00_{0.00}$ | $91.94_{0.68}$ | $17.05_{0.02}$ |
| RivaGan | $99.68_{0.10}$ | $96.78_{0.22}$ | $52.31_{0.01}$ | $100.00_{0.00}$ | $99.13_{0.22}$ | $26.75_{0.02}$ |
| Tree-Ring | - | - | - | $100.00_{0.00}$ | $98.85_{0.31}$ | $6.71_{0.02}$ |
| Gaussian Shading | $100.00_{0.00}$ | $98.43_{0.04}$ | $88.06_{0.11}$ | $100.00_{0.00}$ | $99.97_{0.04}$ | $99.23_{0.00}$ |
| PRC Watermark | $100.00_{0.00}$ | $93.52_{0.20}$ | $97.69_{0.07}$ | $100.00_{0.00}$ | $87.03_{0.39}$ | $95.38_{0.00}$ |
| Ours | $99.99_{0.01}$ | $95.02_{0.08}$ | $98.12_{0.04}$ | $100.00_{0.00}$ | $97.50_{0.18}$ | $99.83_{0.00}$ |

sample one prompt and generate ten watermarked images per user across 100 distinct users for image-level evaluation, with qualitative examples shown in Figure 3. In Figure 2, we also train a ResNet-18 classifier (He et al., 2016) for image-level classification. Tree-Ring and Gaussian Shading are detectable, while PRC Watermark and ours show near-chance detection (50%). Table 1 shows that only PRC Watermark and our method match the original in FID, whereas other methods incur distribution shifts. These results support our theoretical analysis in Section 3.3 by showing that watermarked samples are statistically indistinguishable from unwatermarked ones.

**Computational Efficiency.** To demonstrate the advantages of our encryption-free design, we evaluate the embedding and extraction times of latent-based watermarking schemes, with each result averaged over 100 trials. In this comparison, we focus exclusively on the transformation between the watermark and its latent noise representation, excluding any diffusion sampling or inversion procedures. As illustrated in Figure 4, we employ a logarithmic scale on the y-axis for visualization. The extraction time of the PRC Watermark is much higher than that of ours, roughly four orders of magnitude slower on extraction. This difference reflects the computational burden introduced by belief-propagation decoding in the PRC scheme. In contrast, our approach eliminates the need for complex key design, thereby enhancing execution speed, improving computational efficiency.

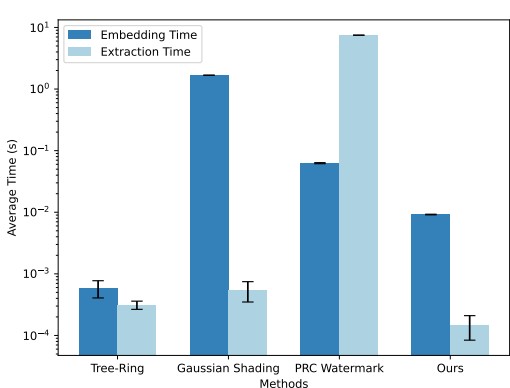

Figure 4: The watermark embedding and extraction time for different latent-based schemes. Here, the Y-axis is on a logarithmic scale.

**Tracing Accuracy.** In Table 2, we evaluate tracing accuracy under varied conditions. "Clean" refers to PNG storage, "Post-Processing" reports common post-processing distortions, and "Adversarial" refers to attacks from (Jiang et al., 2023) (See Appendix F.4 for details). Compared to lossy schemes, our method achieves comparable accuracy above 95% in both Clean and Post-Processing settings. We introduce a tunable parameter $s$, which entails a slight robustness trade-off relative to Gaussian Shading. Under Adversarial conditions, however, the accuracy of lossy schemes degrades sharply, as their embeddings enable effective classifiers to be trained for watermark detection, which can then be adversarially attacked. In contrast, lossless schemes demonstrate clear superiority: our method improves accuracy by more than 10%, consistent with the theoretical analysis in Appendix E.

**Comparison with PRC Watermark.** In Table 2 and Figure 5, we compare our method with PRC Watermark under varied distortions. Our method consistently achieves higher TPR and ACC, with a larger margin at stronger distortions. In addition, Figure 6(a) examines the effect of watermark capacity $l_m$ on tracing accuracy under JPEG–70 compression. As $l_m$ increases, PRC Watermark's decoding performance deteriorates rapidly and fails entirely beyond $l_m = 2000$. In contrast, *Spherical Watermark* sustains high detection rates across the full capacity range. Furthermore, the computational efficiency comparisons show that our embedding and extraction incur significantly lower overhead than PRC Watermark, with extraction being about four orders of magnitude faster. These results confirm the superior robustness of our method.

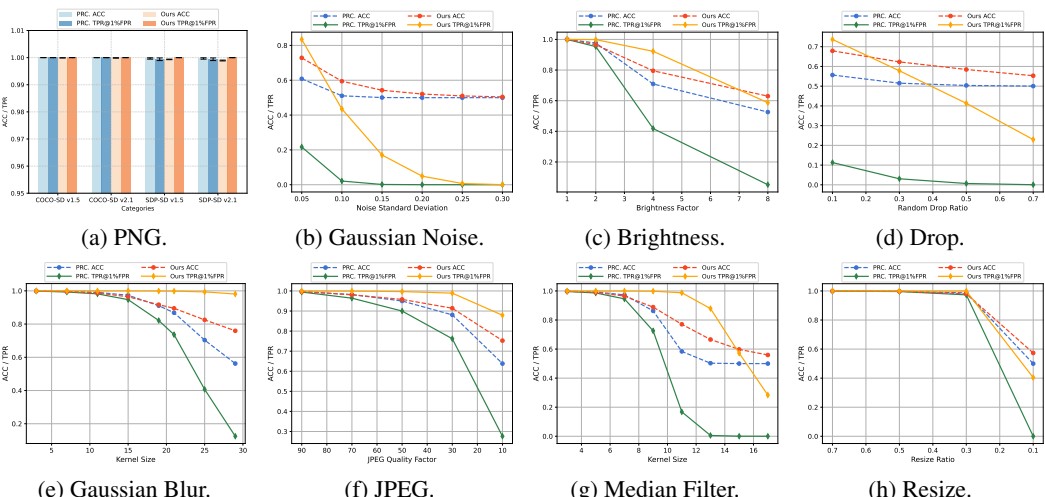

Figure 5: ACC and TPR values under Attacks, averaged over two datasets and two models.

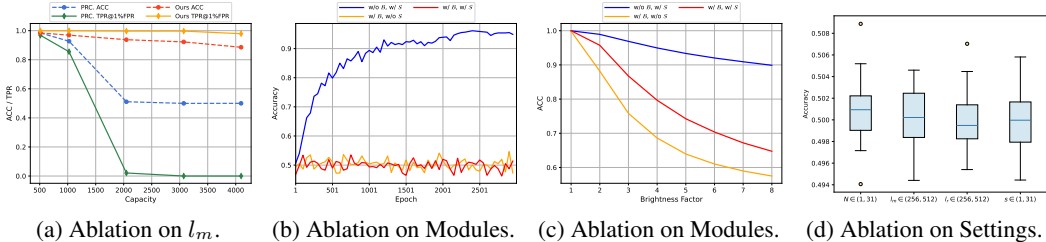

(a) Ablation on $l_m$. (b) Ablation on Modules. (c) Ablation on Modules. (d) Ablation on Settings.

Figure 6: Ablation Study. (a) ACC comparisons under different capacities, (b) and (c) Undetectability and robustness of module ablation, (d) Undetectability analysis of varied parameter settings. Here, (a-c) are all conducted on the COCO dataset with SD v2.1.

### 4.3 ABLATION EXPERIMENTS

**Ablation on Modules.** In our ablation study, we isolate the effects of each module. In one variant, we omit the spherical mapping $\mathcal{S}$ and substitute the Gaussian Shading transform; in another, we skip the binary embedding $\mathcal{B}$ and apply only the spherical mapping to $\mathbf{x}$. We then evaluate both latent-level undetectability and tracing accuracy. In Figure 6(b), omitting the binary embedding makes the latent noise trivially distinguishable. Figure 6(c) shows that robustness under brightness adjustment drops dramatically without spherical mapping. These results confirm that binary embedding enforces independence, while spherical mapping is essential for restoring robustness. A rigorous analysis of why our orthogonal rotation design achieves optimal robustness is provided in Appendix D.

**Ablation on Parameters.** We further investigate the sensitivity of our method to hyperparameters: the watermark length $l_m$, the padding length $l_r$, the row sparsity parameter $s$, and the repetition count $N$. In Figure 6(d), we vary these parameters and train a latent-level classifier to evaluate their effect. The results show that classification accuracy remains near 50%, indicating that parameter changes do not impair undetectability. As $s$ increases, each watermark bit depends on more paddings, making errors more likely to propagate and amplify. Similarly, reducing $N$ decreases redundancy for majority-vote correction. Thus, both larger $s$ and smaller $N$ reduce accuracy by design, a trend also confirmed by the experimental results in Table 3. In addition, Figure 6(a) shows that *Spherical Watermark* maintains high detection rates across all watermark capacities under JPEG–70 compression.

**Ablation on Diffusion Sampling Settings.** We conduct ablation studies on the COCO dataset using the SD v2.1 model to assess the sensitivity of our method to diffusion sampling configurations. Table 4 compares watermark extraction accuracy under various attacks across three ODE solvers: DDIM (Song et al., 2021a), PNDM (Liu et al., 2022), and DPM-Solver++ (Lu et al., 2025). Settings of each attack type are provided in Appendix F.5. We then investigate the role of generation and

Table 3: Ablation of parameters $s$ and $N$ on TPR under different attack. Dataset: COCO. Model: SD v2.1. Case 1: Gaussian Blur, kernel size = 9. Case 2: JPEG-70. Case 3: Brightness, factor = 2.

| Case | sparsity parameter $s$ | | | | repetition count $N$ | | | |
|------|------|------|------|------|------|------|------|------|
|      | 1 | 2 | 3 | 4 | 1 | 11 | 21 | 31 |
| 1 | $100.00_{0.00}$ | $99.84_{0.12}$ | $99.08_{0.25}$ | $97.58_{0.34}$ | $99.40_{0.14}$ | $99.96_{0.05}$ | $99.94_{0.05}$ | $100.00_{0.00}$ |
| 2 | $99.94_{0.05}$ | $99.46_{0.26}$ | $96.64_{0.42}$ | $92.38_{0.74}$ | $98.08_{0.32}$ | $99.86_{0.14}$ | $99.92_{0.12}$ | $99.94_{0.05}$ |
| 3 | $99.72_{0.12}$ | $98.00_{0.21}$ | $93.12_{0.53}$ | $83.68_{0.71}$ | $95.10_{0.45}$ | $99.50_{0.24}$ | $99.68_{0.19}$ | $99.72_{0.12}$ |

Table 4: Ablation results of extraction accuracy on ODE solvers under post-processing perturbations.

| Solver | Post-processing Perturbations | | | | | |
|--------|------|------|------|------|------|------|
|        | PNG | Brightness | Gaussian Blur | Median Filter | JPEG | Resize |
| DDIM | $99.98_{0.01}$ | $96.06_{0.23}$ | $99.43_{0.02}$ | $99.20_{0.03}$ | $98.39_{0.16}$ | $99.85_{0.01}$ |
| PNDM | $99.98_{0.01}$ | $96.17_{0.23}$ | $99.40_{0.02}$ | $99.15_{0.03}$ | $98.41_{0.15}$ | $99.84_{0.01}$ |
| DPM-Solver++ | $99.98_{0.01}$ | $96.02_{0.26}$ | $99.44_{0.01}$ | $99.21_{0.03}$ | $98.40_{0.15}$ | $99.85_{0.01}$ |

Table 5: Ablation results of extraction accuracy on sampling timesteps.

| Generation Timesteps | Inversion Timesteps | | | | |
|--------|------|------|------|------|------|
|        | 10 | 20 | 30 | 40 | 50 |
| 10 | $99.85_{0.88}$ | $99.92_{0.71}$ | $99.95_{0.57}$ | $99.95_{0.58}$ | $99.95_{0.60}$ |
| 20 | $99.96_{0.52}$ | $99.97_{0.48}$ | $99.98_{0.35}$ | $99.98_{0.36}$ | $99.98_{0.38}$ |
| 30 | $99.97_{0.28}$ | $99.99_{0.18}$ | $99.99_{0.16}$ | $99.99_{0.12}$ | $99.99_{0.12}$ |
| 40 | $99.97_{0.56}$ | $99.98_{0.52}$ | $99.98_{0.48}$ | $99.98_{0.47}$ | $99.98_{0.48}$ |
| 50 | $99.97_{0.36}$ | $99.98_{0.36}$ | $99.98_{0.29}$ | $99.99_{0.29}$ | $99.99_{0.26}$ |

inversion timesteps under PNG storage, as summarized in Table 5. Results show that neither the choice of ODE solver nor the variation in timestep schedules introduces meaningful degradation. The minor numerical discrepancies caused by switching solvers or adjusting timesteps are effectively absorbed by the inherent redundancy of our spherical mapping, which provides robustness against moderate inversion inaccuracies. Further quantitative analysis is provided in Appendix F.5.

## 5 DISCUSSION AND LIMITATIONS

Our Gaussian-noise guarantee depends on spherical 3-design definition. While watermarked and random noise are empirically indistinguishable, higher-order moments may deviate from the true prior. Extremely strong inversion-breaking attacks (e.g., perturbations targeting the VAE encoder or ODE solver) can still compromise recovery. We provide an extended analysis of our method against re-generation and editing attacks in Appendix F.2, showing that the proposed approach retains robustness in these scenarios. Nevertheless, our primary focus is on tracing the origin of maliciously generated content. Since editing and forgery may involve different adversarial, such cases are outside our scope. Overall, our method can generalize to any generative model with a Gaussian prior and invertible mappings, with detailed analysis in Appendix F.1.

## 6 CONCLUSION

In this paper, we introduce Spherical Watermark, a novel watermarking framework for image generation that requires no modifications to the diffusion model. Our key innovation is the binary embedding and spherical mapping module that converts binary watermark bits into Gaussian noise input. Watermarked latent inputs are provably and empirically indistinguishable from a standard Gaussian prior. Additionally, we eliminate per-image key management while delivering superior robustness under realistic distortions. Extensive experiments demonstrate that our method outperforms existing schemes in terms of undetectability, traceability, and computational efficiency.

## ACKNOWLEDGMENTS

We sincerely thank the anonymous reviewers and area chairs for their constructive comments and suggestions. This work was supported by the National Natural Science Foundation of China (Grant 62572125) and the Natural Science Foundation of Shanghai (Grant 25ZR1401019).

## ETHICS STATEMENT

We have carefully reviewed and adhered to the ICLR Code of Ethics throughout the development of this work. We have ensured that our methodologies and findings are transparent, reproducible, and free from discrimination, bias, or unfairness concerns. Any potential ethical concerns have been carefully considered, and we encourage responsible use of our contributions in future research.

## REPRODUCIBILITY STATEMENT

We are committed to ensuring the reproducibility of our results. All theoretical claims are supported with complete proofs provided in Appendix C, Appendix D, and Appendix E. For empirical studies, we specify datasets, model architectures, hyperparameters, and implementation details in Section 4, with additional information in Appendix F. The source code is included in the supplementary material, with a README file that clearly documents the execution steps. All experiments are repeated five times, and we report the mean and standard deviation to mitigate randomness and measurement error.

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

## APPENDIX

In the supplement, Appendix A provides a statement on the use of large language models (LLMs) during the preparation. Appendix B presents the technical details of baselines. Appendix C provides the theoretical proofs for the theorem and lemmas introduced in Section 3.3 of the main paper. Appendix D examines the rotation operation within the proposed spherical mapping module, demonstrating its contribution to robustness and showing that it constitutes a near-optimal solution under some lossy channels. Appendix E provides our theoretical analysis of the importance of losslessness, showing that it plays a crucial role in resisting adversarial attacks. Finally, Appendix F supplies additional experimental details and results that further validate the effectiveness of our method.

## A  USE OF LARGE LANGUAGE MODELS

We used large language models (LLMs) to assist with language polishing and improving the clarity of exposition. All technical content, including theoretical claims, proofs, algorithms, and experiments, was developed entirely by the authors.

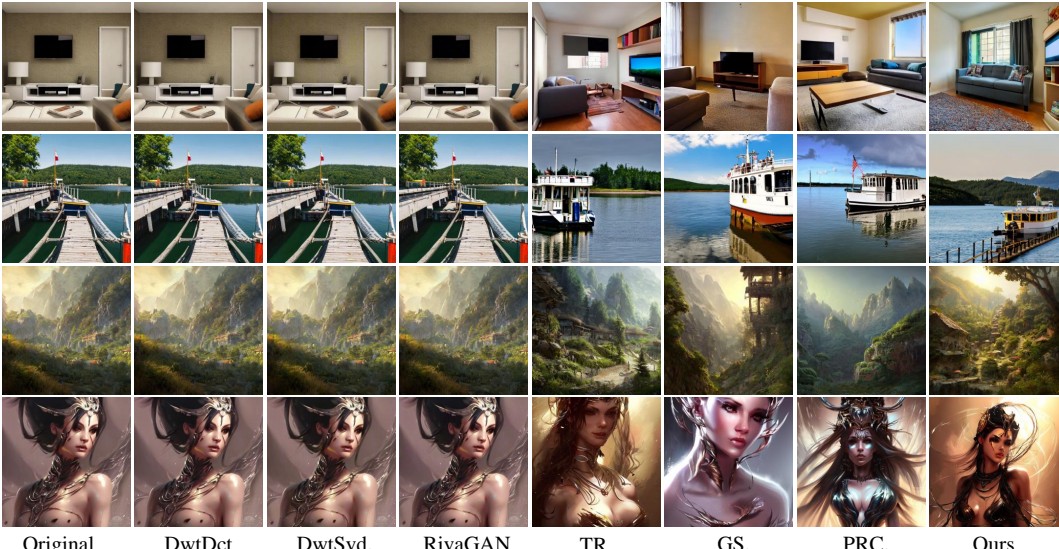

Figure 7: Examples of watermarked images. Generated images from top to bottom: COCO-SD v1.5, COCO-SD v2.1, SDP-SD v1.5, SDP-SD v2.1. Here, "DwtSvd." refers to "DwtDctSvd", "TR." to "Tree-Ring", "GS." to "Gaussian Shading", and "PRC." to "PRC Watermark".

## B    Overview of Baseline Methods

This section systematically reviews and analyzes several previously proposed watermarking schemes evaluated in the experiments. These methods range from traditional signal processing techniques to state-of-the-art latent-based schemes, reflecting the evolution and diversity of watermarking methodologies. We present several examples of watermarked images in Figure 7.

**DwtDct.** DwtDct (Al-Haj, 2007) is a hybrid watermarking technique that integrates Discrete Wavelet Transform (DWT) and Discrete Cosine Transform (DCT). It performs multi-level wavelet decomposition on the carrier image, extracts the low-frequency sub-band to find stable regions, and then applies block-wise DCT to embed the watermark into mid-frequency coefficients. The method balances quality and robustness using DWT for structure retention and DCT for compression resistance.

**DwtDctSvd.** DwtDctSvd (Navas et al., 2008) is an improved watermarking scheme based on DwtDct, enhanced by incorporating Singular Value Decomposition (SVD) to boost security and stability. The method performs DWT on the carrier image, selects specific sub-bands for block-wise DCT, and then performs SVD on each block to embed the watermark into the singular value matrix. It resists geometric transformations and signal processing attacks.

**RivaGAN.** RivaGAN (Zhang et al., 2019) is a watermarking framework based on Generative Adversarial Networks (GANs), using an encoder–decoder architecture for adaptive watermark embedding and extraction. The generator embeds the watermark covertly, while the discriminator and extraction network are trained together to enhance attack resistance. Adversarial training enables adaptability to diverse attacks, ensuring strong robustness.

**Tree-Ring.** The Tree-Ring (Wen et al., 2023) watermarking scheme embeds a specific ring-shaped pattern, referred to as a key into the Fourier space of the initial noise vector in the diffusion model. This subtle modification influences the entire image generation process during sampling, in a way that is imperceptible to humans. By leveraging the properties of the Fourier domain, the watermark exhibits invariance to common attacks.

**Gaussian Shading.** Gaussian Shading (Yang et al., 2024) is a provably secure watermarking scheme. This method utilizes repetition codes and stream encryption to transform the watermark information $\mathbf{m}$ into a binary bit sequence $\mathbf{s} \in \{0, 1\}^n$ that follows a uniformly random distribution, where each bit $s_i \sim \mathrm{Bernoulli}\left(\frac{1}{2}\right), i \in (0, 1, ..., n)$. Then, through a truncated sampling scheme, the sequence $\mathbf{s}$ is mapped into a latent representation $\mathbf{x}$ that conforms to a standard Gaussian distribution. This

process can be formally expressed as:

$$x_i = (2s_i - 1) \cdot |z|, \; z \in \mathcal{N}(0, 1). \tag{14}$$

Here, $z$ is randomly sampled in each invocation, following a standard one-dimensional Gaussian distribution. However, it requires storing a unique key for each image to preserve the original distribution, which introduces substantial management overhead. Moreover, the truncated sampling scheme is suboptimal and fails to reach the theoretical upper bound of error correction, as rigorously analyzed in Appendix D.

**PRC Watermark.** The PRC Watermark (Gunn et al., 2025) embeds information into the latent representation of diffusion models by using pseudorandom error-correcting codes (PRC) (Christ & Gunn, 2024) instead of stream cipher techniques. While their approach eliminates the need to store a separate key for each image, it introduces heavyweight cryptographic complexity. The use of encoding and belief-propagation decoding (Pearl, 2014) incurs substantial computational cost and latency during watermark embedding and extraction. Moreover, their inherent uncertainty leads to them being prone to hitting a performance ceiling, failing to recover the watermark under certain lossy conditions.

## C  Proofs of Theorems and Lemmas in Section 3.3

**Lemma 3.1.** *If $\mathbf{m}$ and $\mathbf{r}$ consist of independent* $\mathrm{Bernoulli}(\frac{1}{2})$ *bits, then for $\mathbf{z}^{(1)}$ in Eq. 11, we have* $z_i^{(1)} \sim \mathrm{Bernoulli}(\frac{1}{2})$ *for every $i \in \{1, \dots, l_x\}$, and $\mathbf{z}^{(1)}$ is both 2-wise and 3-wise independent.*

*Proof.* In $\mathbb{F}_2$, linear independence of Boolean linear forms is equivalent to their statistical independence. We form the combined variable

$$\mathbf{e} = \begin{bmatrix} m_1, m_2, \dots, m_{l_m}, r_1, r_2, \dots, r_{l_r} \end{bmatrix}^\top, \tag{15}$$

which can be viewed as the concatenation of $\mathbf{m}$ and $\mathbf{r}$, with all entries mutually independent. We then construct the binary matrix

$$\mathbf{Q} = \begin{pmatrix} \begin{matrix} \mathbf{I}_{l_m} \\ \mathbf{I}_{l_m} \\ \vdots \\ \underbrace{\mathbf{I}_{l_m}}_{N \text{ copies}} \end{matrix} & \mathbf{R} \\ \mathbf{0}_{l_r \times l_m} & \mathbf{I}_{l_r} \end{pmatrix} \in \{0, 1\}^{(N \times l_m + l_r) \times (l_m + l_r)}, \tag{16}$$

so that the encoding $\mathbf{z}^{(1)} = \mathbf{T}\,\mathbf{x}$ of $\mathbb{F}_2$ in Eq. 11 of the main paper can be compactly re-written as $\mathbf{z}^{(1)} = \mathbf{Q}\,\mathbf{e}$. Since each row of $\mathbf{Q}$ is a nonzero vector in $\mathbb{F}_2^{l_m + l_r}$, we can write

$$z_i^{(1)} = \langle \mathbf{q}_i, \mathbf{e} \rangle_{\mathbb{F}_2} = \bigoplus_{j=1}^{l_m + l_r} q_{i,j}\, e_j, i \in \{1, \dots, l_x\}, \tag{17}$$

where $\mathbf{q}_i$ is the $i$th row of $\mathbf{Q}$, $q_{i,j}$ represents its $j$th entry. $\langle \cdot \rangle_{\mathbb{F}_2}$ denotes the binary inner product, $e_j$ is the $j$th entry of $\mathbf{e}$, and $\bigoplus$ refers to the bitwise XOR. Because the bits $e_j$ are independent and each satisfies $\Pr[e_j = 1] = \frac{1}{2}$, any nontrivial XOR of them remains unbiased, hence

$$\Pr\big[z_i^{(1)} = 1\big] = \Pr\big[z_i^{(1)} = 0\big] = \tfrac{1}{2}, \tag{18}$$

i.e. $z_i^{(1)} \sim \mathrm{Bernoulli}(\frac{1}{2})$ for all $i$.

In this setting, verifying independence of $\mathbf{z}^{(1)}$ reduces to a simple rank condition on $\mathbf{Q}$. Specifically:

> $\mathbf{z}^{(1)}$ *is $k$-wise independent if and only if every $k$-row submatrix of $\mathbf{Q}$ has full row rank.*

Equivalently, for any choice of indices $i_1, \ldots, i_k$,

$$\mathrm{rank}_{\mathbb{F}_2}\big(\mathbf{Q}_{\{i_1, \ldots, i_k\}}\big) \;=\; k, \tag{19}$$

which ensures that the corresponding outputs $z_{i_1}, \ldots, z_{i_k}$ are linearly independent, and hence statistically independent. By Algorithm 1 of the main paper, any two distinct rows of $\mathbf{Q}$ are not identical. Consequently, every 2-row submatrix of $\mathbf{Q}$ has full row rank. It follows that the corresponding output bits are linearly independent, and hence $\mathbf{z}^{(1)}$ is 2-wise independent. Assume, for the sake of contradiction, that $\mathbf{z}^{(1)}$ is not 3-wise independent. Then there exist distinct indices $i_1, i_2, i_3$ ($i_1 < i_2 < i_3$) and coefficients $a, b, c \in \mathbb{F}_2$, not all zero, such that

$$a\, z^{(1)}_{i_1} \;\oplus\; b\, z^{(1)}_{i_2} \;\oplus\; c\, z^{(1)}_{i_3} \;=\; 0. \tag{20}$$

However, since $\mathbf{z}^{(1)}$ is already 2-wise independent, no nontrivial relation can hold among any two bits. For example, if $c = 0$, then we have $a\, z^{(1)}_{i_1} \oplus b\, z^{(1)}_{i_2} = 0$ with $(a, b) \neq (0, 0)$, which contradicts 2-wise independence. Therefore, in the above three-term combination, $a$, $b$ and $c$ must be 1; otherwise we would contradict 2-wise independence. Hence,

$$z^{(1)}_{i_1} \oplus z^{(1)}_{i_2} \oplus z^{(1)}_{i_3} = 0 \;\implies\; \mathbf{q}_{i_1} \oplus \mathbf{q}_{i_2} \oplus \mathbf{q}_{i_3} = \mathbf{0}. \tag{21}$$

The leftmost $l_m$ columns of $\mathbf{Q}$ can sum to zero only in two cases:

- All three indices lie in the bottom block: $i_1$, $i_2$, $i_3 > N \times l_m$. Then in the last $l_r$ columns their rows sum to a nonzero vector, contradicting $\mathbf{q}_{i_1} \oplus \mathbf{q}_{i_2} \oplus \mathbf{q}_{i_3} = \mathbf{0}$.

- Two indices lie in the top block but align on repeated identity rows: $i_1$, $i_2 \leq N \times l_m$, $|i_1 - i_2| \bmod l_m = 0$ and $i_3 > N \times l_m$. In the $\mathbf{R}$ block constructed by Algorithm 1 of the main paper, the positions of the 1–entries in rows $i_1$ and $i_2$ are guaranteed to be disjoint, so their sum over the last $l_r$ columns is nonzero, again a contradiction.

In either case we reach a contradiction. Therefore no nontrivial three-term relation exists, and $\mathbf{z}^{(1)}$ is 3-wise independent. $\qquad\square$

**Definition 3.1** (Spherical $t$-Design). *A finite set of points $X = \{x_1, \ldots, x_N\} \subset S^{n-1}$ on the unit sphere in $\mathbb{R}^n$ is called a* spherical $t$-design *if, for every real polynomial $f$ of total degree at most $t$,*

$$\frac{1}{N} \sum_{x \in X} f(x) \;=\; \frac{1}{|S^{n-1}|} \int_{S^{n-1}} f(x)\, d\sigma(x), \tag{22}$$

*where $d\sigma$ is the uniform surface measure on $S^{n-1}$, and $|S^{n-1}|$ denotes the total surface area of the unit $(n-1)$-sphere.*

Equivalently, $X$ is a $t$-design if and only if it *matches all moments* of the uniform distribution on the sphere up to degree $t$. In the following theorem, we show that the finite point set sampled from $\mathbf{z}^{(2)}$ constitutes a spherical 3-design.

**Theorem 3.2.** *$\mathbf{z}^{(2)}$ satisfies that each $z^{(2)}_i$ takes values $\pm\frac{1}{\sqrt{l_x}}$ with $\Pr[z_i = +\frac{1}{\sqrt{l_x}}] = \Pr[z_i = -\frac{1}{\sqrt{l_x}}] = \frac{1}{2}$, $i \in (1, \cdots, l_x)$; $\mathbf{z}^{(2)}$ is 2-wise and 3-wise independent. Then the finite set of $\mathbf{z}^{(2)}$ on the unit sphere $S^{l_x - 1}$ is a spherical 3–design.*

*Proof.* $\mathbf{z}^{(1)} \in \{0, 1\}^{l_x}$ is 2-wise and 3-wise independent with $\Pr[z^{(1)}_i = 1] = \Pr[z^{(1)}_i = 0] = \frac{1}{2}$, $i, \in \{1, \ldots, l_x\}$. Define

$$\mathbf{v} = 2\,\mathbf{z}^{(1)} - \mathbf{1}, \; \mathbf{z}^{(2)} = \frac{\mathbf{v}}{\|\mathbf{v}\|_2}. \tag{23}$$

Since $z^{(1)}_i \sim \mathrm{Bernoulli}(\frac{1}{2})$, we have $v_i = 2z^{(1)}_i - 1 \in \{\pm 1\}$ with $\Pr[v_i = 1] = \Pr[v_i = -1] = \frac{1}{2}$. Because $\|\mathbf{v}\|_2 = \sqrt{l_x}$, it follows that

$$z^{(2)}_i \;=\; \frac{v_i}{\sqrt{l_x}} \;\in\; \left\{\pm\frac{1}{\sqrt{l_x}}\right\}, \tag{24}$$

and $\Pr[z_i^{(2)} = +\frac{1}{\sqrt{l_x}}] = \Pr[z_i^{(2)} = -\frac{1}{\sqrt{l_x}}] = \frac{1}{2}$. Note that the transformation from $\mathbf{z}^{(1)}$ to $\mathbf{z}^{(2)}$ consists solely of a affine shift followed by a normalization. Neither step mixes entries across different coordinates: each $z_i^{(2)}$ depends only on the corresponding $z_i^{(1)}$. Moreover, both operations are invertible. Hence any independence property of $\mathbf{z}^{(1)}$ is preserved exactly in $\mathbf{z}^{(2)}$, and no new dependencies are introduced. We can derive that $\mathbf{z}^{(2)}$ is also 2-wise and 3-wise independent.

By the polynomial-averages characterization of spherical designs (Delsarte–Goethals–Seidel 1977 (Delsarte et al., 1977), Theorem 5.3; Bannai & Bannai 2009 (Bannai & Bannai, 2009), Theorem 2.2), it suffices to check that $\mathbf{z}^{(2)}$ matches the uniform sphere's moments for every monomial of total degree $\leq 3$. Let $\mathbf{U} = (U_1, \ldots, U_{l_x})$ be a random vector drawn uniformly from the unit sphere $S^{l_x - 1}$. By symmetry,

$$\mathbb{E}[U_i] = 0, \quad \mathbb{E}[U_i^2] = \frac{1}{l_x}, \quad \mathbb{E}[U_i U_j] = 0 \quad (i \neq j), \tag{25}$$

and all mixed moments of total degree up to three that are odd in any coordinate equal zero. We list all cases:

**Degree 1.**

$$\mathbb{E}[z_i^{(2)}] = \frac{1}{2}\left(\frac{1}{\sqrt{l_x}} + \left(-\frac{1}{\sqrt{l_x}}\right)\right) = 0, \tag{26}$$

matching the sphere's $\mathbb{E}[U_i] = 0$.

**Degree 2.**

$$\mathbb{E}[(z_i^{(2)})^2] = \left(\frac{1}{\sqrt{l_x}}\right)^2 = \frac{1}{l_x}, \quad \mathbb{E}[z_i^{(2)} z_j^{(2)}] = \mathbb{E}[z_i^{(2)}]\,\mathbb{E}[z_j^{(2)}] = 0 \quad (i \neq j), \tag{27}$$

which agrees with the uniform-sphere values $\mathbb{E}[U_i^2] = 1/l_x$, $\mathbb{E}[U_i U_j] = 0$.

**Degree 3.**

$$\mathbb{E}[(z_i^{(2)})^3] = \left(\frac{1}{\sqrt{l_x}}\right)^3 \left(\frac{1}{2} - \frac{1}{2}\right) = 0, \tag{28}$$

$$\mathbb{E}[(z_i^{(2)})^2 z_j^{(2)}] = \mathbb{E}[(z_i^{(2)})^2]\,\mathbb{E}[z_j^{(2)}] = \frac{1}{l_x} \cdot 0 = 0 \quad (i \neq j), \tag{29}$$

$$\mathbb{E}[z_i^{(2)} z_j^{(2)} z_k^{(2)}] = \mathbb{E}[z_i^{(2)}]\,\mathbb{E}[z_j^{(2)}]\,\mathbb{E}[z_k^{(2)}] = 0 \quad (\text{distinct } i, j, k). \tag{30}$$

These all match the sphere's vanishing of odd moments up to degree 3. Since every polynomial of degree $\leq 3$ is a linear combination of these monomials, $\mathbf{z}^{(2)}$ reproduces exactly the uniform-sphere averages through degree 3. Hence, the finite set of $\mathbf{z}^{(2)}$ is a spherical 3–design. $\qquad\square$

**Lemma 3.3.** *Let $\mathbf{z}^{(2)} \in S^{l_x - 1}$ be a spherical 3–design. If we apply a fixed orthogonal rotation $\mathbf{z}^{(3)} = \mathbf{C}\,\mathbf{z}^{(2)}$, then $\mathbf{z}^{(3)}$ is also a spherical 3–design. For each coordinate $z_i^{(3)}$, one has $\mathbb{E}[z_i] = 0$ and $\mathbb{E}[z_i^2] = 1/l_x$, and as $l_x \to \infty$, the marginal law of $z_i^{(3)}$ converges to $\mathcal{N}(0, 1/l_x)$.*

*Proof.* Since the point set

$$X = \{\mathbf{z}^{(2)}\} \subset S^{l_x - 1} \tag{31}$$

is a spherical 3–design, it integrates exactly all polynomials of total degree $\leq 3$. Hence for any polynomial $f : \mathbb{R}^{l_x} \to \mathbb{R}$ with $\deg f \leq 3$,

$$\frac{1}{|X|} \sum_{\mathbf{z}^{(2)} \in X} f(\mathbf{z}^{(2)}) = \frac{1}{|S^{l_x - 1}|} \int_{S^{l_x - 1}} f(x)\, d\sigma(x). \tag{32}$$

Define the rotated set

$$X' = \{\mathbf{z}^{(3)} : \mathbf{z}^{(3)} = \mathbf{C}\,\mathbf{z}^{(2)}, \ \mathbf{z}^{(2)} \in X\}. \tag{33}$$

Since orthogonal rotations preserve the spherical surface (Haar) measure,

$$\int_{S^{l_x - 1}} f(\mathbf{C}x)\, d\sigma(x) = \int_{S^{l_x - 1}} f(x)\, d\sigma(x). \tag{34}$$

Because $f(\mathbf{C}x)$ is still a polynomial of degree $\leq 3$, the design property of $\{\mathbf{z}^{(2)}\}$ gives

$$
\begin{aligned}
\frac{1}{|X'|} \sum_{\mathbf{z}^{(3)} \in X'} f\big(\mathbf{z}^{(3)}\big) &= \frac{1}{|X|} \sum_{\mathbf{z}^{(2)} \in X} f\big(\mathbf{C}\,\mathbf{z}^{(2)}\big) \\
&= \frac{1}{|S^{l_x-1}|} \int_{S^{l_x-1}} f\big(\mathbf{C}x\big)\, d\sigma(x) = \frac{1}{|S^{l_x-1}|} \int_{S^{l_x-1}} f(x)\, d\sigma(x).
\end{aligned}
\tag{35}
$$

Therefore the rotated set $X'$ also constitutes a spherical 3–design.

Recall that from the spherical 3–design property in the proof of Theorem 3.2 (or equivalently from the degree–1 and degree–2 moment matching) we have for the original vector $\mathbf{z}^{(2)} = (z_1^{(2)}, \ldots, z_{l_x}^{(2)})^\top$:

$$
\mathbb{E}\big[z_j^{(2)}\big] = 0, \quad \mathbb{E}\big[z_j^{(2)} z_k^{(2)}\big] = \frac{1}{l_x}\, \delta_{jk}, \quad j, k = 1, \ldots, l_x.
\tag{36}
$$

the symbol $\delta_{jk}$ denotes the *Kronecker delta*, which is defined by

$$
\delta_{jk} = \begin{cases} 1, & j = k, \\ 0, & j \neq k. \end{cases}
\tag{37}
$$

Let $\mathbf{z}^{(3)} = \mathbf{C}\,\mathbf{z}^{(2)}$ be the rotated vector, so that

$$
z_i^{(3)} = \sum_{j=1}^{l_x} C_{ij}\, z_j^{(2)}.
\tag{38}
$$

Here, $C_{ij}$ represents the $i$th row, $j$th column entry of $\mathbf{C}$. By linearity of expectation and the above moment identities,

$$
\mathbb{E}\big[z_i^{(3)}\big] = \sum_{j=1}^{l_x} C_{ij}\, \mathbb{E}\big[z_j^{(2)}\big] = 0,
\tag{39}
$$

and

$$
\begin{aligned}
\mathbb{E}\big[(z_i^{(3)})^2\big] &= \mathbb{E}\Big[\sum_j^{l_x} \sum_k^{l_x} C_{ij} C_{ik}\, z_j^{(2)}\, z_k^{(2)}\Big] \\
&= \sum_j^{l_x} \sum_k^{l_x} C_{ij} C_{ik}\, \mathbb{E}[z_j^{(2)} z_k^{(2)}] = \frac{1}{l_x} \sum_{j=1}^{l_x} C_{ij}^2 = \frac{1}{l_x},
\end{aligned}
\tag{40}
$$

The classical central limit theorem for independent summands is not directly applicable here since $\mathbf{z}^{(2)}$ may exhibit local dependence. Instead, we construct the dependency graph according to Stein's method (Ross, 2011; Chen & Shao, 2004). For the rotated item $z_i^{(3)}$ derived by Eq. 38, we build a dependency graph with $l_x$ vertices, where vertex $j$ corresponds to $C_{ij}\, z_j^{(2)}$. Following the generation procedure, each entry of $\mathbf{z}^{(2)}$ is given by

$$
z_j^{(2)} = \frac{2\langle \mathbf{q}_j, \mathbf{e}\rangle_{\mathbb{F}_2} - 1}{\sqrt{l_x}},
\tag{41}
$$

where $\mathbf{q}_j$ denotes the $j$-th row of the binary matrix $\mathbf{Q}$ (see Eq. 16), and $\mathbf{e}$ is defined by Eq. 15. We connect two distinct vertices $j_1 \neq j_2$ by an edge if and only if $\mathbf{q}_{j_1}$ and $\mathbf{q}_{j_2}$ share at least one common index at which both entries equal 1, i.e., $\mathrm{supp}(\mathbf{q}_{j_1}) \cap \mathrm{supp}(\mathbf{q}_{j_2}) \neq \varnothing$, where $\mathrm{supp}(\mathbf{q}_j) := \{k | q_{j,k} = 1\}$ denotes the support set of indices. Equivalently, an edge exists when $z_{j_1}^{(2)}$ and $z_{j_2}^{(2)}$ share at least one underlying source bit from $\mathbf{m}$ or $\mathbf{r}$. Moreover, $\mathbb{E}[(z_i^{(3)})^2] = 1/l_x$, and we can derive a standard normal-approximation bound for locally dependent sums. By Theorem 3.6 in (Ross, 2011), if the degree of the dependency graph is bounded by the maximum degree $D$, then for the Wasserstein distance $d_W$ of $\sqrt{l_x} z_i^{(3)}$ and $\mathcal{N}(0, 1)$, it satisfies,

$$
d_W \leq D^2 \sum_{j=1}^{l_x} \mathbb{E}\Big[\big|\sqrt{l_x}\, C_{ij} z_j^{(2)}\big|^3\Big] + \sqrt{\frac{28}{\pi}}\, D^{3/2} \Big(\sum_{j=1}^{l_x} \mathbb{E}\Big[\big(\sqrt{l_x}\, C_{ij} z_j^{(2)}\big)^4\Big]\Big)^{1/2}.
\tag{42}
$$

Obviously, $\big|\sqrt{l_x}\, z_j^{(2)}\big| = 1$, so the right-hand side reduces to terms involving $|C_{ij}|$. Next, we verify that our dependency graph satisfies the conditions above.

- Due to the constraint imposed by Algorithm 1, the maximum degree $D$ is uniquely determined by the embedding matrix $\mathbf{T}$ and satisfies $D \leq N + l_m - 1$. Since $N$ and $l_m$ are preset constants, we can treat $D$ as a constant.

- The entries $C_{ij}$ come from the orthogonal matrix $\mathbf{C}$, which is sampled from the Haar-uniform measure according to (Mezzadri, 2006). For a typical Haar-random $\mathbf{C}$ and sufficiently large $l_x$, the row-wise empirical averages concentrate around their Haar expectations in probability over the draw of $\mathbf{C}$. For any fixed $(i, j)$, the squared entry $|C_{ij}|^2$ follows the Beta distribution $\mathrm{Beta}\left(\frac{1}{2}, \frac{l_x - 1}{2}\right)$ (Wikipedia contributors, 2025; Livan et al., 2018). In particular, the third and fourth absolute moments are,

$$\mathbb{E}\left[|C_{ij}|^3\right] \;=\; \frac{\Gamma\left(\frac{l_x}{2}\right)}{\sqrt{\pi}\,\Gamma\left(\frac{l_x+3}{2}\right)} \;=\; \Theta\left(l_x^{-3/2}\right), \quad \mathbb{E}\left[|C_{ij}|^4\right] \;=\; \frac{\Gamma\left(\frac{5}{2}\right)\Gamma\left(\frac{l_x}{2}\right)}{\sqrt{\pi}\,\Gamma\left(\frac{l_x+4}{2}\right)} \;=\; \Theta\left(l_x^{-2}\right),$$
(43)

where $\Gamma(\cdot)$ denotes the Gamma function, and $\Theta(\cdot)$ is the standard asymptotic order notation.

Therefore, ignoring constant factors, the distributional discrepancy in Eq. 42 can be simplified as,

$$d_W \;=\; O\left(D^2\, l_x^{-1/2}\right) \;+\; O\left(D^{3/2}\, l_x^{-1/2}\right) \;=\; O\left(l_x^{-1/2}\right),$$
(44)

where $O(\cdot)$ denotes the standard Big-$O$ notation. As $l_x \to \infty$, we have $d_W \to 0$, which implies that the marginal distribution of $z_i^{(3)}$ converges to $\mathcal{N}(0, 1/l_x)$.

This completes the proof. $\qquad\square$

**Lemma 3.4.** *Let $\mathbf{x} \sim \mathcal{N}(\mathbf{0}, \mathbf{I}_n)$ be a standard multivariate normal vector in $\mathbb{R}^n$. Then $\mathbf{x}$ admits a polar decomposition of the form*

$$\mathbf{x} = r \cdot \mathbf{u},$$
(45)

*where $r^2 \sim \chi^2(n)$, and $\mathbf{u}$ is uniformly distributed on the unit sphere $S^{n-1}$. Furthermore, $r$ and $\mathbf{u}$ are statistically independent. Conversely, if $r^2 \sim \chi^2(n)$, $\mathbf{u}$ is uniformly distributed on $S^{n-1}$, and $r \perp \mathbf{u}$, then the product $\mathbf{x} = r \cdot \mathbf{u}$ follows a standard multivariate normal distribution, i.e., $\mathbf{x} \sim \mathcal{N}(\mathbf{0}, \mathbf{I}_n)$.*

*Proof.* In the sequel, we denote the Euclidean norm $\|\cdot\|_2$ simply by $\|\cdot\|$. We begin by demonstrating that any random vector $\mathbf{x}$ drawn from a standard multivariate normal distribution can be decomposed into a radial component $r$ and a directional component $\mathbf{u}$. We refer to a classical result on spherical distributions (see Theorem 1.5.6 in (Muirhead, 2009)), which states,

> **Theorem.** If $\mathbf{x}$ has an $n$-variate spherical distribution with $\Pr\left[\mathbf{x} = \mathbf{0}\right] = 0$, and we define the radial part as $r = \|\mathbf{x}\| = (\mathbf{x}^\top \mathbf{x})^{1/2}$ and the normalized direction as $T(\mathbf{x}) = \|\mathbf{X}\|^{-1}\mathbf{X}$, then $T(\mathbf{X})$ is uniformly distributed on the unit sphere $S^{n-1}$, and $T(\mathbf{X})$ and $r$ are statistically independent.

Since the multivariate normal distribution is continuous, we have $\Pr\left[\mathbf{x} = \mathbf{0}\right] = 0$. Moreover, the standard normal distribution is rotationally invariant; that is, for any orthogonal matrix $\mathbf{C} \in \mathbb{R}^{n \times n}$, we have $\mathbf{C}\mathbf{x} \stackrel{d}{=} \mathbf{x}$ (The notation $\stackrel{d}{=}$ denotes equality in distribution). Therefore, $\mathbf{x}$ is spherically distributed in $\mathbb{R}^n$, and it satisfies all the conditions required by the theorem. By applying the above theorem, we decompose $\mathbf{x}$ into a radial part and a direction as follows:

$$r = \|\mathbf{x}\| = \left(\sum_{i=1}^{n} x_i^2\right)^{1/2}, \quad \mathbf{u} = T(\mathbf{x}) = \frac{\mathbf{x}}{\|\mathbf{x}\|}.$$
(46)

Since each $x_i \sim \mathcal{N}(0, 1)$ and the components are independent, it follows that $r^2 \sim \chi^2(n)$. The direction $\mathbf{u}$ is uniformly distributed over the unit sphere $S^{n-1}$, and $r$ and $\mathbf{u}$ are independent by the theorem. This establishes the sufficiency: any standard multivariate normal vector $\mathbf{x} \sim \mathcal{N}(\mathbf{0}, \mathbf{I}_n)$ can be decomposed into an independent product of a $\chi_n$-distributed magnitude and a uniformly distributed direction.

Conversely, suppose we independently sample $\mathbf{u} \sim \text{Unif}(S^{n-1})$ and $r \sim \chi_n$, and define $\mathbf{z} = r \cdot \mathbf{u}$. Here $\text{Unif}(S^{n-1})$ represents the uniform distribution one the unit sphere $S^{n-1}$. Then, using the change-of-variables formula in polar coordinates, the probability density of $\mathbf{z}$ in $\mathbb{R}^n$ is given by:

$$f_{\mathbf{z}}(\mathbf{z}) = f_r(\|\mathbf{z}\|) \cdot f_{\mathbf{u}}\left(\frac{\mathbf{z}}{\|\mathbf{z}\|}\right) \cdot \|\mathbf{z}\|^{-(n-1)}, \tag{47}$$

where $f_r(\cdot)$ and $f_{\mathbf{u}}(\cdot)$ are the densities of $\chi_n$ and the uniform sphere distribution, respectively. Substituting in the known densities and simplifying, we recover the standard multivariate normal density:

$$f_{\mathbf{z}}(\mathbf{z}) = \frac{1}{(2\pi)^{n/2}} \exp\left(-\frac{\|\mathbf{z}\|^2}{2}\right). \tag{48}$$

Therefore, $\mathbf{z} \sim \mathcal{N}(\mathbf{0}, \mathbf{I}_n)$. $\qquad\qquad\qquad\qquad\qquad\qquad\qquad\qquad\qquad\qquad\qquad\qquad\quad \square$

## D  ROLE OF ROTATION IN ENHANCING ROBUSTNESS

In the ablation experiments of the main paper, we observed that replacing the spherical mapping module with the provably secure transformation introduced in Gaussian Shading results in a degradation of robustness. In this section, we analyze the underlying reasons for this behavior. *Note: To avoid notational conflicts, this section is self-contained and uses its own symbols.*

First, we compare the provably secure mapping used in Gaussian Shading with the spherical mapping in our scheme. Let the watermark-bearing bit-string be denoted by the vector $\mathbf{s} \in \{0,1\}^n$. We first convert $\mathbf{s}$ into a $\{\pm 1\}^n$ vector via

$$\mathbf{s} \leftarrow 2\,\mathbf{s} - \mathbf{1}. \tag{49}$$

In the Gaussian Shading pipeline, one then samples a Gaussian noise vector $\mathbf{z} \sim \mathcal{N}(\mathbf{0}, \mathbf{I}_n)$ (of the same dimension as $\mathbf{s}$) and forms the watermarked Gaussian noise $\mathbf{x}$ by element-wise multiplication:

$$\mathbf{x} = |\mathbf{z}| \odot \mathbf{s}. \tag{50}$$

Here, $|\cdot|$ represents the element-wise absolute value operator. By contrast, our scheme applies a fixed orthogonal rotation:

$$\mathbf{x} = r\mathbf{C}\,\mathbf{s}, \tag{51}$$

where $\mathbf{C} \in \mathbb{R}^{n \times n}$ is an orthogonal matrix satisfying $\mathbf{C}^\top \mathbf{C} = \mathbf{I}_n$. Here, $\mathbf{I}_n$ is the identity matrix of size $n \times n$. The scalar $r$ is defined as

$$r = \frac{\sqrt{q}}{\sqrt{n}}, \quad q \sim \chi_n^2, \tag{52}$$

i.e., $q$ follows a chi-square distribution with $n$ degrees of freedom.

**AWGN channel.** We now compare the per-bit extraction accuracy of the two mappings under an additive white Gaussian noise (AWGN) channel. Let the received vector be

$$\mathbf{y} = \mathbf{x} + \boldsymbol{e}, \quad \boldsymbol{e} \sim \mathcal{N}(\mathbf{0}, \sigma^2 \mathbf{I}_n). \tag{53}$$

In Gaussian Shading, each coordinate of $\mathbf{x}$ takes the form

$$x_i = |z_i|\, s_i, \quad z_i \sim \mathcal{N}(0,1), \quad s_i \in \{\pm 1\}. \tag{54}$$

We extract $\hat{s}_i$ by the sign of the noisy sample $y_i = x_i + e_i$. Hence, the per-bit extraction accuracy by Gaussian Shading is

$$P_{\text{GS}} = P\big(\hat{s}_i = s_i\big) = \int_{-\infty}^{0} \phi(x)\, P\big(x + e_i < 0\big)\,\mathrm{d}x + \int_{0}^{\infty} \phi(x)\, P\big(x + e_i > 0\big)\,\mathrm{d}x. \tag{55}$$

Let $\phi$ and $\Phi$ are the standard Gaussian density and cumulative distribution function (CDF). Since

$$P\big(x + e_i > 0\big) = \Phi\big(x/\sigma\big), \quad P\big(x + e_i < 0\big) = 1 - \Phi\big(x/\sigma\big), \tag{56}$$

we get

$$P_{\text{GS}} = \int_{-\infty}^{0} \phi(x)\left[1 - \Phi(x/\sigma)\right]\mathrm{d}x + \int_{0}^{\infty} \phi(x)\,\Phi(x/\sigma)\,\mathrm{d}x = \int_{0}^{\infty} 2\,\phi(x)\,\Phi(x/\sigma)\,\mathrm{d}x. \tag{57}$$

In our spherical mapping scheme,

$$\mathbf{x} = r\,\mathbf{C}\,\mathbf{s}, \quad \mathbf{C}^\top \mathbf{C} = \mathbf{I}_n, \tag{58}$$

Upon reception we compute

$$\mathbf{y}' = \mathbf{C}^\top \mathbf{y} = \mathbf{C}^\top (r\,\mathbf{C}\,\mathbf{s} + \boldsymbol{e}) = r\,\mathbf{s} + \underbrace{\mathbf{C}^\top \boldsymbol{e}}_{\sim\,\mathcal{N}(0,\sigma^2\mathbf{I}_n)}. \tag{59}$$

Extraction is done coordinate-wise by $\hat{s}_i = \mathrm{sign}(y_i')$, the per-bit accuracy of our method is derived by

$$
\begin{aligned}
P_{\mathrm{Ours}} &= P\big(\hat{s}_i = s_i\big) \\
&= P\big(r\,s_i + \eta_i > 0 \mid s_i = +1\big)\,P(s_i = +1) + P\big(r\,s_i + \eta_i < 0 \mid s_i = -1\big)\,P(s_i = -1) \\
&= \tfrac{1}{2}\,P\big(r + \eta_i > 0\big) + \tfrac{1}{2}\,P\big(-r + \eta_i < 0\big) \\
&= \Phi\big(r/\sigma\big), \eta_i \sim \mathcal{N}(0,\sigma^2).
\end{aligned}
\tag{60}
$$

Recall the design of $r$ in Eq. 52, so that $\mathbb{E}[r] = 1$ and $\mathrm{Var}(r) = O(1/n)$. In particular, for large $n$,

$$\mathrm{Var}(r) = \frac{n - \big(\mathbb{E}[\chi_n]\big)^2}{n} \approx \frac{1/2}{n} = \frac{1}{2n} \longrightarrow 0 \quad (n \to \infty). \tag{61}$$

Hence $r$ concentrates around 1 in probability,

$$\mathbb{E}_r\big[\Phi(r/\sigma)\big] \longrightarrow \Phi\big(1/\sigma\big), \; P_{\mathrm{Ours}} \longrightarrow \Phi\big(1/\sigma\big). \tag{62}$$

By definition,

$$P_{\mathrm{GS}} = \int_0^\infty 2\,\phi(x)\,\Phi\big(x/\sigma\big)\,\mathrm{d}x = \mathbb{E}_U\big[\Phi(U/\sigma)\big], \tag{63}$$

where $U = |Z|$ with $Z \sim N(0,1)$ (so $U$ is half-normal on $[0,\infty)$). Since $\Phi$ is strictly concave on $[0,\infty)$ and $\mathbb{E}[U] = \sqrt{2/\pi} < 1$, Jensen's inequality gives

$$P_{\mathrm{GS}} = \mathbb{E}_U\big[\Phi(U/\sigma)\big] \leq \Phi\big(\mathbb{E}[U]/\sigma\big) < \Phi\big(1/\sigma\big) = P_{\mathrm{Ours}}. \tag{64}$$

Hence $P_{\mathrm{GS}} < P_{\mathrm{Ours}}$ for all $\sigma > 0$. In other words, under the AWGN channel, our proposed scheme achieves a higher per-bit extraction accuracy than Gaussian Shading.

**Optimality under Equal-Energy Constraint.** In the sequel, we denote the Euclidean norm $\|\cdot\|_2$ simply by $\|\cdot\|$. Both Gaussian Shading and our mapping embed $\mathbf{s} \in \{\pm 1\}^n$ such that the *total* energy, measured by the squared Euclidean norm of the embedded vector, is on average $n$, yielding a *per-bit* energy of exactly 1. In other words, $\mathbb{E}\big[\mathbf{x_i}^2\big] = 1, i = 1, \ldots, n$. Concretely:

$$\mathbf{x}_{\mathrm{GS}} = |\mathbf{z}| \odot \mathbf{s}, \; \mathbf{z} \sim \mathcal{N}(\mathbf{0}, \mathbf{I}_n), \tag{65}$$

so $\|\mathbf{x}_{\mathrm{GS}}\|^2 = \|\mathbf{z}\|^2$ and $\mathbb{E}(\|\mathbf{x}_{\mathrm{GS}}\|^2) = n$.

$$\mathbf{x}_{\mathrm{Ours}} = r\,\mathbf{C}\,\mathbf{s}, \; \mathbf{C}^\top \mathbf{C} = \mathbf{I}_n, \; \mathbb{E}[r^2] = 1, \tag{66}$$

so $\|\mathbf{x}_{\mathrm{Ours}}\|^2 = r^2\,\|\mathbf{s}\|^2 = r^2 n$ and $\mathbb{E}(\|\mathbf{x}_{\mathrm{Ours}}\|^2) = n$. After de-rotation by $\mathbf{C}^\top$, bit $i$ yields

$$y_i' = r\,s_i + \eta_i, \quad \eta_i \sim \mathcal{N}(0,\sigma^2), \tag{67}$$

so the two hypotheses correspond to symbols $+r$ and $-r$ with distance $\Delta_{\mathrm{Ours}} = 2r \to 2$ in probability as $n \to \infty$. In Gaussian Shading, bit $i$ is observed as

$$y_i = \pm|z_i| + e_i, \quad z_i, e_i \sim \mathcal{N}(0,1), \tag{68}$$

so $\Delta_{\mathrm{GS}} = 2|z_i|$ whose expectation $2\sqrt{2/\pi} < 2$.

It is a classical result in digital communications (Proakis, Digital Communications (Proakis & Salehi, 2008), Section 5.2) and information theory (Cover & Thomas, Elements of Information Theory (Cover, 1999)) that, under equal-energy constraints in any symmetric, memoryless additive-noise channel, larger Euclidean separation between symbols strictly reduces bit-error probability. Since our scheme attains the maximal possible distance of 2 for unit-energy symbols, it is provably optimal among all binary mappings with the same energy constraints.

# E  WHY LOSSLESSNESS MATTERS: AN DETECTOR-DRIVEN OPTIMIZATION ANALYSIS

Our goal is to justify, in a detector-aware threat model, that losslessness (distributional indistinguishability between watermarked and unwatermarked inputs) is more fundamental than merely increasing robustness. If the watermarked distribution leaves any detectable shift from the clean distribution, an adversary can exploit it to mount effective detector-aware attacks; if the two distributions coincide (losslessness), the optimization collapses to blind perturbations with little extra damage.

Consider an input image $\mathbf{x}$, and let $\boldsymbol{\delta}$ denote an adversarial perturbation constrained by $\|\boldsymbol{\delta}\|_2 \le \varepsilon$. We write the attack objective as

$$\min_{\|\boldsymbol{\delta}\|_2 \le \varepsilon} F_{\mathbf{x}}(\boldsymbol{\delta}) = \mathcal{L}(s(\mathbf{x} + \boldsymbol{\delta})) + \tfrac{\lambda}{2}\|\boldsymbol{\delta}\|_2^2, \tag{69}$$

where $s(\cdot) \in \mathbb{R}$ is the detector score, $\mathcal{L}$ is a loss applied to the score, and $\lambda > 0$ controls the trade-off between reducing the score and keeping the perturbation small. The loss $\mathcal{L}$ is chosen to encourage the detector prediction to flip, similar to the binary cross-entropy (BCE) loss used in standard classification. For instance, if $s(\mathbf{x})$ is interpreted as the probability that $\mathbf{x}$ is watermarked, an adversary may apply a BCE-type objective with the target label flipped from "watermarked" to "clean". In this case, the loss explicitly drives the detector to misclassify a watermarked input as unwatermarked. The derivatives $\mathcal{L}'$ of such objectives are uniformly bounded, ensuring that the gradient scaling factor they introduce remains finite.

Next, we derive the optimal perturbation $\boldsymbol{\delta}^\star$ that minimizes $F_{\mathbf{x}}$. A first-order expansion at $\boldsymbol{\delta} = \mathbf{0}$ gives (we use the notation $\langle\cdot,\cdot\rangle$ to denote the Euclidean inner product of vectors),

$$F_{\mathbf{x}}(\boldsymbol{\delta}) \approx F_{\mathbf{x}}(\mathbf{0}) + \langle \mathbf{a}_{\mathbf{x}}, \boldsymbol{\delta} \rangle + \tfrac{\lambda}{2}\|\boldsymbol{\delta}\|_2^2, \quad \mathbf{a}_{\mathbf{x}} := \mathcal{L}'(s(\mathbf{x}))\,\nabla_x s(\mathbf{x}). \tag{70}$$

The optimal perturbation is given by

$$\boldsymbol{\delta}^\star(\mathbf{x}) = -\min\left\{\varepsilon,\ \tfrac{\|\mathbf{a}_{\mathbf{x}}\|_2}{\lambda}\right\} \frac{\mathbf{a}_{\mathbf{x}}}{\|\mathbf{a}_{\mathbf{x}}\|_2}. \tag{71}$$

Substituting $\boldsymbol{\delta}^\star$ into the objective yields the decrease

$$F_{\mathbf{x}}(\boldsymbol{\delta}^\star) - F_{\mathbf{x}}(\mathbf{0}) = -\tfrac{1}{2}\min\left\{\tfrac{\|\mathbf{a}_{\mathbf{x}}\|_2^2}{\lambda},\ 2\varepsilon\|\mathbf{a}_{\mathbf{x}}\|_2 - \lambda\varepsilon^2\right\}. \tag{72}$$

This shows that the decrease is monotone in the gradient magnitude: when $\|\mathbf{a}_{\mathbf{x}}\|_2$ is large, the optimizer exploits the distortion budget and achieves a significant reduction; when $\|\mathbf{a}_{\mathbf{x}}\|_2 \approx 0$, the decrease vanishes and the attack is ineffective.

Next, we will analyze how the lossless and lossy cases differ in terms of the expected gradient energy, and prove that lossless watermarking methods yield smaller values of $\mathbb{E}\|\mathbf{a}_{\mathbf{x}}\|_2^2$, whereas lossy methods result in larger values. Formally, let $P_{\mathrm{wm}}$ and $P_{\mathrm{clean}}$ denote the watermarked and clean distributions with densities $p_{\mathrm{wm}}$ and $p_{\mathrm{clean}}$, respectively. If they differ, the KL divergence is strictly positive. By the Neyman–Pearson lemma, the *optimal* detector takes the form of the log-likelihood ratio,

$$s(\mathbf{x}) = \log \frac{p_{\mathrm{wm}}(\mathbf{x})}{p_{\mathrm{clean}}(\mathbf{x})}. \tag{73}$$

Its gradient satisfies

$$\nabla_{\mathbf{x}} s(\mathbf{x}) = \nabla_{\mathbf{x}} \log p_{\mathrm{wm}}(\mathbf{x}) - \nabla_{\mathbf{x}} \log p_{\mathrm{clean}}(\mathbf{x}), \tag{74}$$

and therefore

$$\mathbb{E}_{P_{\mathrm{wm}}}\|\nabla_{\mathbf{x}} s(\mathbf{x})\|_2^2 = J(P_{\mathrm{wm}}\|P_{\mathrm{clean}}), \tag{75}$$

the Fisher divergence between the two distributions. It is known that

$$J(P_{\mathrm{wm}}\|P_{\mathrm{clean}}) \gtrsim C\,D_{\mathrm{KL}}(P_{\mathrm{wm}}\|P_{\mathrm{clean}}), \tag{76}$$

for some constant $C > 0$. Since $\mathbf{a}_{\mathbf{x}}$ only differs from $\nabla_{\mathbf{x}} s(\mathbf{x})$ by the bounded factor $\mathcal{L}'(s(\mathbf{x}))$, we conclude that

$$\mathbb{E}\|\mathbf{a}_{\mathbf{x}}\|_2^2 \gtrsim C\,D_{\mathrm{KL}}(P_{\mathrm{wm}}\|P_{\mathrm{clean}}). \tag{77}$$

Thus, the KL divergence provides a lower bound on the expected gradient energy. In lossy cases, when $D_{\mathrm{KL}}(P_{\mathrm{wm}}\|P_{\mathrm{clean}}) > 0$, the gradients remain informative and provide actionable descent directions,

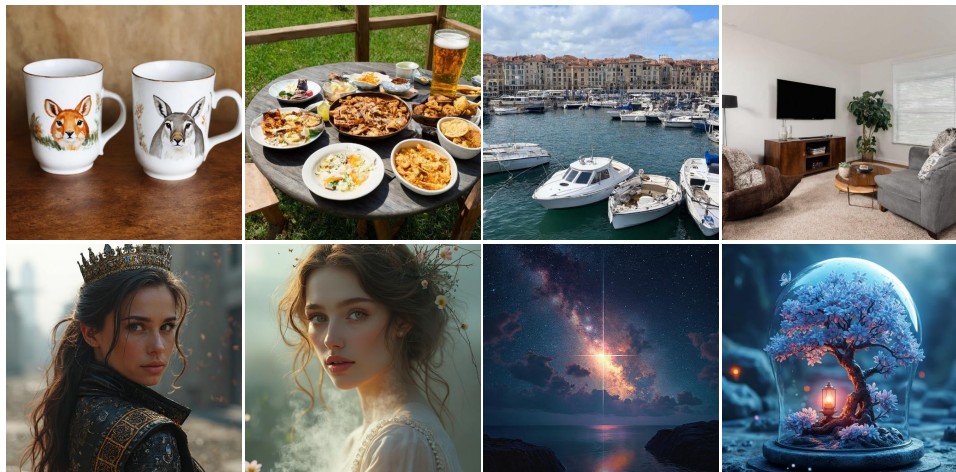

Figure 8: Watermarked images generated by SD v3 and FLUX.1-DEV. Top: COCO dataset with SD v3. Bottom: SDP dataset with FLUX.1-DEV.

Table 6: Extraction performance for different models under post-processing perturbations. The strength of the post-processing perturbations are consistent with Table 4 in the main paper.

| Method | Metrics | Post-processing Perturbations | | | | | |
|---|---|---|---|---|---|---|---|
| | | PNG | Brightness | Gaussian Blur | Median Filter | JPEG | Resize |
| FLUX.1-DEV | ACC | $99.99_{0.01}$ | $98.84_{0.28}$ | $95.42_{0.28}$ | $91.92_{0.35}$ | $88.85_{0.56}$ | $99.10_{0.13}$ |
| FLUX.1-DEV | TPR | $100.00_{0.00}$ | $99.80_{0.40}$ | $100.00_{0.00}$ | $100.00_{0.00}$ | $99.90_{0.20}$ | $100.00_{0.00}$ |
| SD v3 | ACC | $99.99_{0.01}$ | $98.67_{0.11}$ | $99.83_{0.10}$ | $99.74_{0.11}$ | $98.97_{0.10}$ | $99.92_{0.06}$ |
| SD v3 | TPC | $100.00_{0.00}$ | $99.33_{0.24}$ | $100.00_{0.00}$ | $100.00_{0.00}$ | $99.50_{0.00}$ | $100.00_{0.00}$ |

allowing the adversary to exploit the distortion budget and cause substantial degradation. In contrast, in the lossless case $P_{\mathrm{wm}} = P_{\mathrm{clean}}$, we have $\nabla_{\mathbf{x}} s(\mathbf{x}) = \mathbf{0}$, the expected gradient energy collapses, and the optimizer stalls at $\boldsymbol{\delta} \approx \mathbf{0}$. This analysis demonstrates why losslessness is fundamental: only when the watermarked and clean distributions are indistinguishable does the adversary lose the ability to perform adversarial attacks that evade the detection system.

## F  MORE DETAILS AND EXPERIMENTAL RESULTS

### F.1  GENERALIZABILITY

Our method demonstrates strong generalizability and can be readily adapted to other latent-space diffusion architectures, including those based on transformer designs (Vaswani et al., 2017; Dosovitskiy et al., 2021). As an example, we deploy our scheme on Stable Diffusion v3 (SD v3) (Esser et al., 2024) and FLUX.1-DEV (BlackForestLabs, 2024). For SD v3, we use a first-order Euler ODE solver for both generation and inversion, with 50 timesteps for generation and 20 timesteps for inversion. The generated images have a resolution of $1024 \times 1024$ with prompts sampled from the SDP dataset, and each image carries a 512-bit watermark. For FLUX.1-DEV, we adopt RF-Solver (Wang et al., 2024) for both generation

Table 7: Watermark tracing accuracy on Guided Diffusion and Glow.

| Model | $l_m$ | Accuracy (%) |
|---|---|---|
| G-Diffusion | 96 | $98.75_{6.15}$ |
| | 192 | $98.78_{6.00}$ |
| | 384 | $98.75_{5.92}$ |
| Glow | 32 | $99.97_{0.16}$ |
| | 64 | $99.97_{0.20}$ |
| | 128 | $99.73_{0.18}$ |

and inversion, each using 30 timesteps. We evaluate the model on prompts from the COCO dataset. Each generated image is sized $512 \times 512$, with a 512-bit watermark embedded. Figure 8 presents visual examples of watermarked images, which preserve high perceptual fidelity. To further evaluate the adaptability of our watermarking scheme, we measure extraction performance under a variety

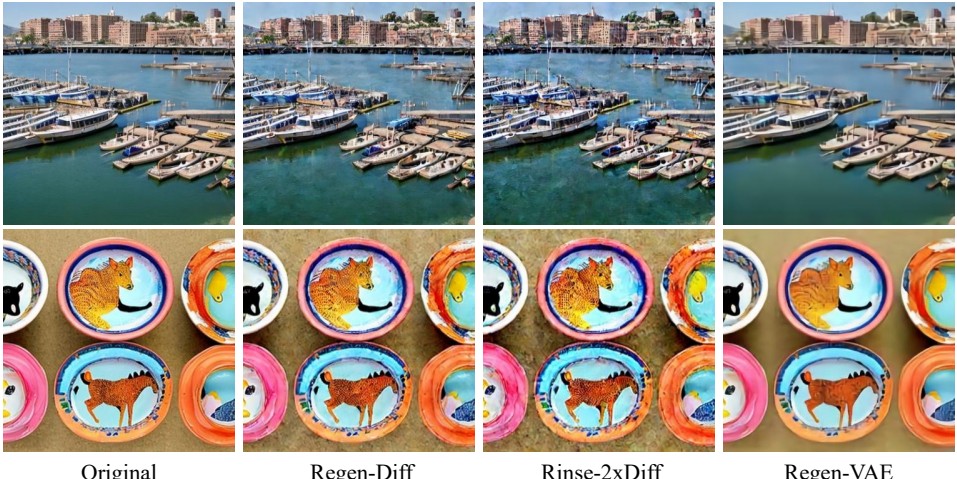

| Original | Regen-Diff | Rinse-2xDiff | Regen-VAE |

Figure 9: Visualized comparison of watermarked images under re-generation.

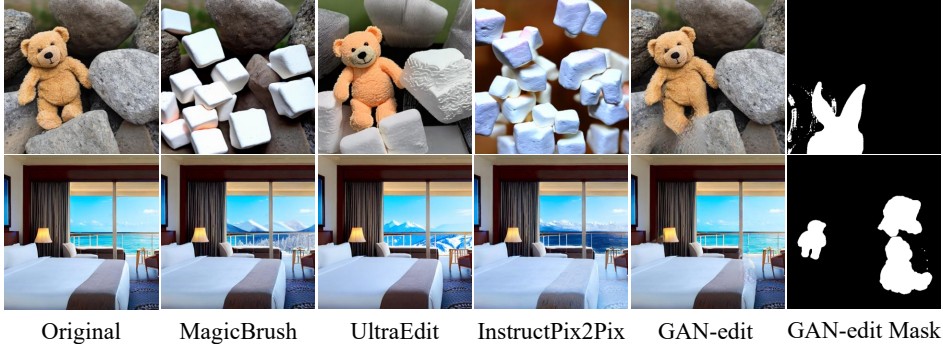

| Original | MagicBrush | UltraEdit | InstructPix2Pix | GAN-edit | GAN-edit Mask |

Figure 10: Visualized comparison of watermarked images under image editing.

of post-processing operations in Table 6. Across these attack settings, the extraction performance remains high, indicating that the proposed scheme adapts well to different latent-space diffusion architectures.

Moreover, our method is not limited to latent-space diffusion architectures. To assess its broader applicability, we apply our scheme to two classic generative models: G-Diffusion (Dhariwal & Nichol, 2021), a pixel-space diffusion model trained on ImageNet (Deng et al., 2009), and Glow (Kingma & Dhariwal, 2018), a flow-based model with invertible transformations trained on the CIFAR-10 dataset (Krizhevsky et al., 2009). We evaluate on 1000 images for each model and report watermark tracing accuracy across multiple watermark lengths (Table 7). In both cases, extraction accuracy under lossless conditions exceeds 98%, demonstrating the effectiveness of our method.

More broadly, our approach applies to any generative model that satisfies two conditions: (1) sampling from a Gaussian prior, and (2) supporting an invertible mapping between the image and noise domains. These conditions are met by a wide range of architectures, including diffusion models (Rombach et al., 2022), normalizing flows (e.g., NICE (Dinh et al., 2014), Glow (Kingma & Dhariwal, 2018)), GANs with inversion (Abdal et al., 2019).

### F.2 ROBUSTNESS AGAINST RE-GENERATION AND EDITING

In Table 8, We evaluate the extraction accuracy of different watermarking methods against image re-generation and image editing attacks. We additionally include TrustMark (Bui et al., 2025) and Robust-Wide (Hu et al., 2024) for comparison, as both methods represent more recent advances in robust watermarking. TrustMark provides a resolution-agnostic pixel-space watermarking framework

Table 8: Extraction accuracy for re-generation and editing attacks. Here, $l_m$ means watermark length, "Robust." refers to "Robust-Wide", and "InstructP2P." refers to "InstructPix2Pix".

| Method | $l_m$ | Re-generation | | | Editing | | | |
|---|---|---|---|---|---|---|---|---|
| | | Regen-Diff | Rinse-2xDiff | Regen-VAE | MagicBrush | UltraEdit | InstructP2P. | GAN-edit |
| DwtDct | 32 | $49.96_{0.73}$ | $49.91_{0.46}$ | $50.27_{0.72}$ | $50.25_{1.28}$ | $49.28_{1.96}$ | $50.56_{1.38}$ | $49.88_{2.21}$ |
| DwtDctSvd | 32 | $50.20_{0.93}$ | $49.69_{0.74}$ | $48.98_{0.58}$ | $49.22_{1.76}$ | $49.62_{1.62}$ | $48.53_{0.88}$ | $48.28_{3.44}$ |
| RivaGan | 32 | $56.77_{0.56}$ | $54.19_{0.53}$ | $50.89_{0.33}$ | $67.31_{1.24}$ | $52.06_{2.33}$ | $60.56_{1.52}$ | $99.84_{0.14}$ |
| TrustMark | 100 | $71.36_{0.46}$ | $59.94_{0.30}$ | $93.87_{0.33}$ | $86.74_{1.85}$ | $68.16_{1.18}$ | $81.68_{3.54}$ | $95.85_{1.34}$ |
| Robust. | 64 | $96.90_{0.18}$ | $93.22_{0.23}$ | $94.15_{0.63}$ | $94.36_{1.54}$ | $80.09_{0.68}$ | $96.44_{1.10}$ | $98.33_{0.61}$ |
| PRC | 100 | $99.26_{0.41}$ | $94.22_{0.88}$ | $75.91_{1.25}$ | $94.14_{2.33}$ | $81.53_{3.09}$ | $83.53_{7.82}$ | $100.00_{0.00}$ |
| Ours | 100 | $99.63_{0.17}$ | $97.70_{0.37}$ | $87.48_{0.45}$ | $93.96_{2.94}$ | $86.11_{1.61}$ | $92.53_{2.38}$ | $100.00_{0.00}$ |

Table 9: PSNR and SSIM for traditional watermarking methods. Model: SD v1.5.

| Method | COCO | | SDP | |
|---|---|---|---|---|
| | PSNR | SSIM | PSNR | SSIM |
| DwtDct | $38.0899_{0.1516}$ | $0.9998_{0.0000}$ | $38.4649_{0.1114}$ | $0.9994_{0.0001}$ |
| DwtDctSvd | $38.0736_{0.1596}$ | $0.9998_{0.0000}$ | $38.9261_{0.2489}$ | $0.9999_{0.0000}$ |
| RivaGan | $40.5620_{0.0045}$ | $0.9968_{0.0001}$ | $40.8294_{0.0713}$ | $0.9905_{0.0017}$ |

with superior robustness. Robust-Wide focuses on robustness against instruction-driven image editing, and we evaluate its 64-bit configuration following the original implementation. For a fair comparison, we adjust the watermark length of our method to 100 bits.

For re-generation attacks, we adopt the WAVES benchmark (Ding et al., 2024), which contains three re-generation attacks: Regen-Diff, Rinse-2xDiff, and Regen-VAE. Specific attacked examples are shown in the Figure 9. The experimental results on the COCO dataset using SD v2.1 are reported in Table 8. Our method achieves the highest accuracy under both Regen-Diff and Rinse-2xDiff, demonstrating superior robustness to diffusion-based re-generation attacks. In the Regen-VAE setting, TrustMark and Robust-Wide obtain slightly higher accuracy, while our method still outperforms the lossless PRC Watermark across all attack scenarios.

For image editing and forgery, we employ the W-Bench benchmark (Lu et al., 2024). This evaluation covers a range of manipulations, including MagicBrush (Zhang et al., 2023), UltraEdit (Zhao et al., 2024), InstructPix2Pix (Brooks et al., 2023). We also introduce GAN-edit (Ma et al., 2022), a GAN-based inpainting method. Figure 10 presents examples of the applied image editing attacks. The experiments are conducted on the W-Bench dataset using the SD v2.1. As shown in the Table 8, our scheme delivers high extraction accuracy across diverse editing operations, staying above 85% on average across all attack scenarios. These results indicate that our robustness against image editing is broadly comparable to existing state-of-the-art watermarking methods.

### F.3 UNDETECTABILITY AND IMAGE QUALITY

In this section, we present additional experiments on both undetectability and image quality assessment. First, Tables 9 and 10 report the Peak Signal-to-Noise Ratio (PSNR) and Structural Similarity Index Measure (SSIM) (Wang et al., 2004) of traditional watermarking methods. These results indicate that conventional approaches tend to degrade image quality, with an average PSNR of 39 dB. To further evaluate the perceptual quality of watermarked images generated by latent-based methods, we incorporate the Inception Score (IS) (Salimans et al., 2016) as an additional metric, as shown in Table 11. Although the IS is not sensitive to distribution shifts, we still observe that lossy watermarking schemes tend to yield lower IS scores compared to lossless ones. This observation aligns with our theoretical expectation that lossy watermarking inherently induces distributional deviations.

For undetectability, we compare the classification performance of trained detectors on watermarked versus unwatermarked images in Figure 11, and further illustrate the ability to distinguish watermarked images generated by different keys in Figure 12. The PRC Watermark and our method exhibit strong

Table 10: PSNR and SSIM for traditional watermarking methods. Model: SD v2.1.

| Method | COCO | | SDP | |
|---|---|---|---|---|
| | PSNR | SSIM | PSNR | SSIM |
| DwtDct | $38.5041_{0.1484}$ | $0.9999_{0.0000}$ | $38.0640_{0.2543}$ | $0.9999_{0.0000}$ |
| DwtDctSvd | $38.6557_{0.1862}$ | $0.9999_{0.0000}$ | $37.9792_{0.2627}$ | $0.9999_{0.0000}$ |
| RivaGan | $40.5628_{0.0065}$ | $0.9968_{0.0001}$ | $40.5111_{0.0064}$ | $0.9968_{0.0001}$ |

Table 11: IS value for different watermarking methods. Higher IS indicates higher image quality.

| Method | COCO | | SDP | |
|---|---|---|---|---|
| | SD v1.5 | SD v2.1 | SD v1.5 | SD v2.1 |
| Original | $29.3546_{1.8784}$ | $28.8780_{1.7803}$ | $11.5965_{0.7757}$ | $10.2936_{0.7868}$ |
| DwtDct | $29.2378_{1.7822}$ | $28.4645_{1.4584}$ | $10.9270_{0.7261}$ | $10.1593_{0.7254}$ |
| DwtDctSvd | $29.4093_{1.7302}$ | $28.6121_{1.4145}$ | $11.0862_{0.7460}$ | $10.2226_{0.6963}$ |
| RivaGan | $29.2398_{1.7400}$ | $28.2529_{1.3353}$ | $10.9988_{0.7410}$ | $10.1668_{0.7217}$ |
| Tree-Ring | $29.3528_{0.3373}$ | $28.8102_{1.3819}$ | $11.5670_{0.2605}$ | $10.0701_{0.7270}$ |
| Gaussian Shading | $29.0868_{0.2818}$ | $28.4530_{1.6960}$ | $11.4438_{0.5970}$ | $10.1322_{0.1784}$ |
| PRC Watermark | $29.4244_{1.5163}$ | $28.9931_{1.5686}$ | $11.6887_{0.6781}$ | $10.2102_{0.8206}$ |
| Ours | $29.5521_{2.2552}$ | $28.9781_{1.7562}$ | $11.6628_{0.6790}$ | $10.2672_{0.7218}$ |

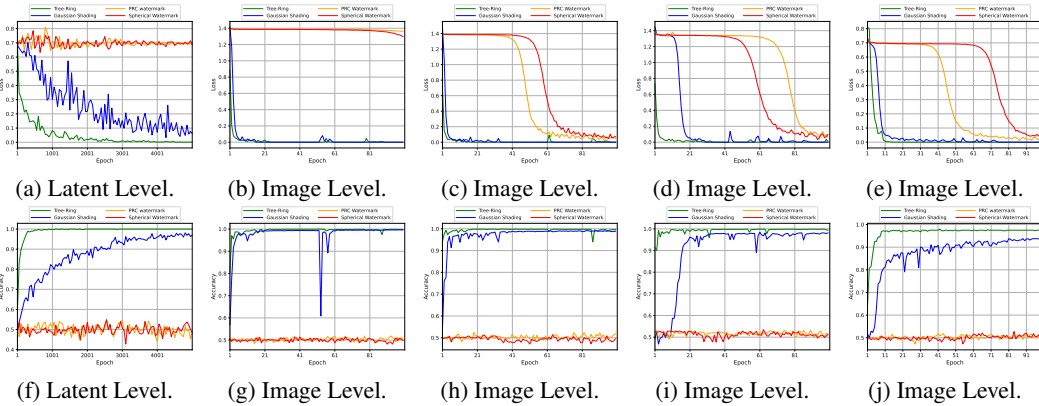

(a) Latent Level.  (b) Image Level.  (c) Image Level.  (d) Image Level.  (e) Image Level.

(f) Latent Level.  (g) Image Level.  (h) Image Level.  (i) Image Level.  (j) Image Level.

Figure 11: Training loss and test classification accuracy for watermark and unwatermarked samples classification. (a)(f) on the latent level. (b)(g) on the COCO dataset with SD v1.5. (c)(h) on the COCO dataset with SD v2.1. (d)(i) on the SDP dataset with SD v1.5. (e)(j) on the SDP dataset with SD v2.1. Top: Training loss. Bottom: Test classification accuracy.

resistance to classification, maintaining strong indistinguishability in both latent and image levels. In contrast, the Tree-Ring and Gaussian Shading methods, which rely on fixed key patterns, introduce significant and consistent artifacts. These artifacts are easily captured by the detectors, resulting in classification accuracies of 100% and 98%, respectively. This indicates that such methods lack robustness against adversarial classification and are highly susceptible to reverse-engineering attacks.

## F.4 TRACING ACCURACY

We evaluate tracing performance in Table 12, 13, and 14, comparing the two diffusion model variants under clean, post-processing, and adversarial conditions using the COCO dataset and SDP dataset. As mentioned in the main paper, "Clean" refers to PNG storage, while "Post-Processing" reports the average performance under a range of common image perturbations. These post-processing perturbations include Additive White Gaussian Noise with a standard deviation of 0.05, Brightness Adjustment with a factor of 2, Random Drop with a drop ratio of 0.1, Gaussian Blur with a kernel size of 9, JPEG compression with a quality factor of 70 (QF=70), Median Filter with a kernel size of

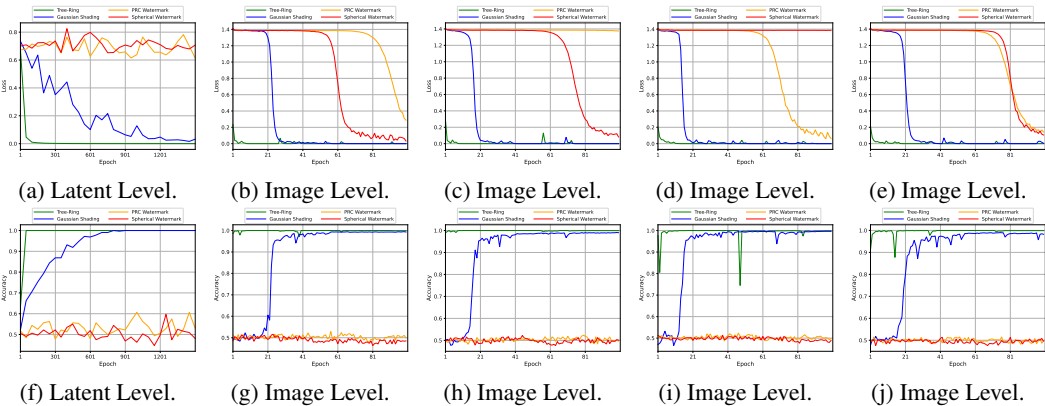

(a) Latent Level.    (b) Image Level.    (c) Image Level.    (d) Image Level.    (e) Image Level.

(f) Latent Level.    (g) Image Level.    (h) Image Level.    (i) Image Level.    (j) Image Level.

Figure 12: Training loss and test classification accuracy on watermarked images generated by different keys. (a)(f) on the latent level. (b)(g) on the COCO dataset with SD v1.5. (c)(h) on the COCO dataset with SD v2.1. (d)(i) on the SDP dataset with SD v1.5. (e)(j) on the SDP dataset with SD v2.1. Top: Training loss. Bottom: Test classification accuracy.

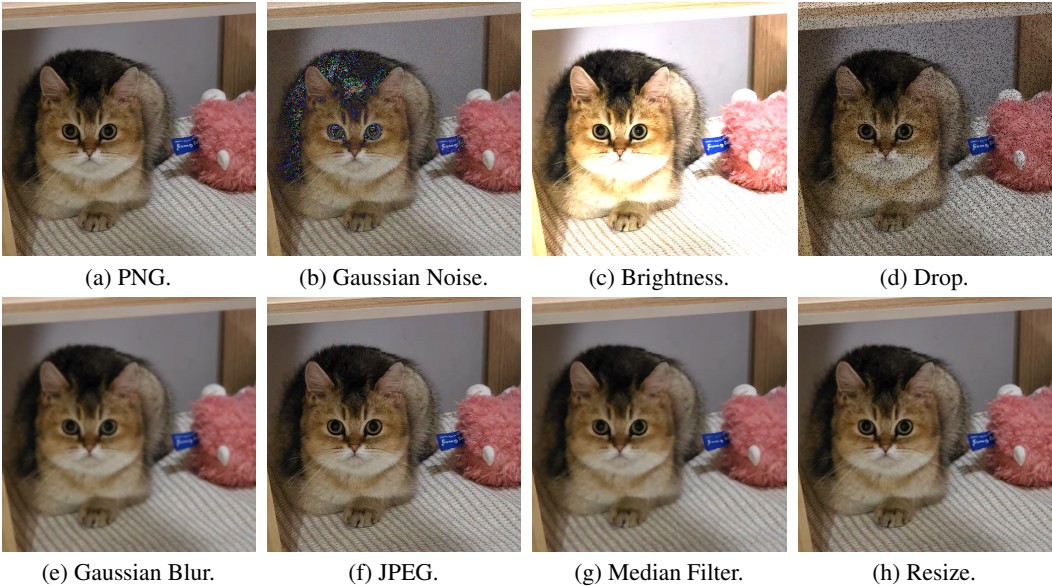

(a) PNG.    (b) Gaussian Noise.    (c) Brightness.    (d) Drop.

(e) Gaussian Blur.    (f) JPEG.    (g) Median Filter.    (h) Resize.

Figure 13: Images under common attacks. (a) PNG image. (b) Additive White Gaussian Noise, $\mu = 0, \sigma = 0.05$. (c) Brightness, factor=2. (d) Drop, drop ratio=0.1. (e) Gaussian Blur, kernel size=9. (f) JPEG, QF=70. (g) Median Filter, kernel size=5. (f) 50% Resize.

5, and 50% Resize (followed by restoration to the original dimensions). Representative examples of these perturbations are shown in Figure 13.

For the Adversarial conditions, we adopt the adversarial attack WEvade (Jiang et al., 2023). In the White-box setting, we use WEvade-W-I with the parameters: maximum pixel perturbation $rb = 10$, gradient learning rate $\alpha = 2$, and error rate $\epsilon = 0.01$. For the Black-box setting, we employ WEvade-B-S, which relies only on binary output predictions, and additionally adopt the JPEG compression with a quality factor of 50 (QF=50) from WEvade-B-Q, before applying the same default parameters as in the White-box setting. A pre-trained ResNet-18 is employed as a surrogate model for generating the adversarial perturbations. The final results are obtained by averaging over the two settings.

In Table 12, 13, and 14, our method achieves over 95% ACC and TPR across Clean and Post-Processing settings. Under Adversarial conditions, the accuracy of lossy schemes drops sharply,

Table 12: Comparison results on ACC and TPR. Dataset: COCO dataset. Model: SD v1.5. "Post." refers to "Post-Processing" and "Adv." refers to "Adversarial".

| Method | Metrics | | | | | |
|---|---|---|---|---|---|---|
| | ACC (Clean) | ACC (Post.) | ACC (Adv.) | TPR (Clean) | TPR (Post.) | TPR (Adv.) |
| DwtDct | $88.83_{0.74}$ | $64.34_{0.73}$ | $49.22_{0.00}$ | $92.70_{1.03}$ | $51.46_{1.72}$ | $15.80_{0.01}$ |
| DwtDctSvd | $100.00_{0.01}$ | $92.86_{0.26}$ | $48.71_{0.00}$ | $100.00_{0.00}$ | $91.49_{0.71}$ | $18.45_{0.01}$ |
| RivaGan | $99.55_{0.15}$ | $96.59_{0.31}$ | $52.78_{0.01}$ | $100.00_{0.00}$ | $99.13_{0.25}$ | $27.85_{0.03}$ |
| Tree-Ring | - | - | - | $94.50_{3.96}$ | $93.98_{4.24}$ | $13.72_{0.07}$ |
| Gaussian Shading | $100.00_{0.00}$ | $98.55_{0.04}$ | $89.42_{0.11}$ | $100.00_{0.00}$ | $99.95_{0.03}$ | $99.30_{0.00}$ |
| PRC Watermark | $100.00_{0.00}$ | $93.75_{0.14}$ | $98.30_{0.07}$ | $100.00_{0.00}$ | $87.51_{0.28}$ | $96.61_{0.01}$ |
| Ours | $99.99_{0.00}$ | $95.27_{0.11}$ | $98.50_{0.03}$ | $100.00_{0.00}$ | $97.57_{0.18}$ | $99.95_{0.00}$ |

Table 13: Comparison results on ACC and TPR. Dataset: SDP dataset. Model: SD v2.1. "Post." refers to "Post-Processing" and "Adv." refers to "Adversarial".

| Method | Metrics | | | | | |
|---|---|---|---|---|---|---|
| | ACC (Clean) | ACC (Post.) | ACC (Adv.) | TPR (Clean) | TPR (Post.) | TPR (Adv.) |
| DwtDct | $83.07_{0.74}$ | $63.14_{0.70}$ | $48.83_{0.01}$ | $87.80_{1.25}$ | $52.77_{2.23}$ | $15.60_{0.03}$ |
| DwtDctSvd | $99.97_{0.02}$ | $92.50_{0.26}$ | $49.40_{0.01}$ | $100.00_{0.00}$ | $89.46_{0.41}$ | $20.20_{0.03}$ |
| RivaGan | $99.06_{0.21}$ | $95.87_{0.38}$ | $50.77_{0.00}$ | $100.00_{0.00}$ | $98.94_{0.36}$ | $22.40_{0.03}$ |
| Tree-Ring | - | - | - | $100.00_{0.00}$ | $98.59_{0.35}$ | $2.84_{0.01}$ |
| Gaussian Shading | $99.99_{0.00}$ | $98.47_{0.04}$ | $84.50_{0.10}$ | $100.00_{0.00}$ | $99.96_{0.03}$ | $99.69_{0.00}$ |
| PRC Watermark | $99.97_{0.03}$ | $93.59_{0.18}$ | $96.19_{0.10}$ | $99.94_{0.05}$ | $87.19_{0.36}$ | $92.39_{0.00}$ |
| Ours | $99.89_{0.02}$ | $94.92_{0.09}$ | $96.94_{0.05}$ | $100.00_{0.00}$ | $97.69_{0.18}$ | $99.87_{0.00}$ |

Table 14: Comparison results on ACC and TPR. Dataset: SDP dataset. Model: SD v1.5. "Post." refers to "Post-Processing" and "Adv." refers to "Adversarial".

| Method | Metrics | | | | | |
|---|---|---|---|---|---|---|
| | ACC (Clean) | ACC (Post.) | ACC (Adv.) | TPR (Clean) | TPR (Post.) | TPR (Adv.) |
| DwtDct | $84.91_{1.55}$ | $63.23_{0.87}$ | $49.55_{0.01}$ | $90.50_{3.03}$ | $51.54_{2.54}$ | $16.20_{0.02}$ |
| DwtDctSvd | $100.00_{0.00}$ | $92.48_{0.35}$ | $49.24_{0.00}$ | $100.00_{0.00}$ | $89.46_{0.59}$ | $17.85_{0.01}$ |
| RivaGan | $99.32_{0.11}$ | $96.34_{0.30}$ | $51.48_{0.01}$ | $100.00_{0.00}$ | $98.97_{0.23}$ | $23.95_{0.04}$ |
| Tree-Ring | - | - | - | $100.00_{0.00}$ | $99.10_{0.39}$ | $4.67_{0.03}$ |
| Gaussian Shading | $99.99_{0.01}$ | $98.49_{0.06}$ | $85.80_{0.10}$ | $100.00_{0.00}$ | $99.95_{0.02}$ | $99.81_{0.00}$ |
| PRC Watermark | $99.97_{0.02}$ | $93.58_{0.20}$ | $97.28_{0.09}$ | $99.94_{0.05}$ | $87.16_{0.39}$ | $94.58_{0.00}$ |
| Ours | $99.93_{0.01}$ | $95.01_{0.09}$ | $97.53_{0.05}$ | $100.00_{0.00}$ | $97.63_{0.19}$ | $99.81_{0.00}$ |

Table 15: Ablation of parameters $s$ and $N$ on TPR under different attacks. Dataset: COCO. Model: SD v1.5. Case 1: Gaussian Blur, kernel size = 9. Case 2: JPEG-70. Case 3: Brightness, factor = 2.

| Case | sparsity parameter $s$ | | | | repetition count $N$ | | | |
|---|---|---|---|---|---|---|---|---|
| | 1 | 2 | 3 | 4 | 1 | 11 | 21 | 31 |
| 1 | $99.96_{0.05}$ | $99.80_{0.06}$ | $99.34_{0.26}$ | $98.42_{0.29}$ | $99.58_{0.15}$ | $99.98_{0.04}$ | $99.90_{0.06}$ | $99.96_{0.05}$ |
| 2 | $99.98_{0.04}$ | $99.60_{0.23}$ | $98.72_{0.32}$ | $94.98_{0.86}$ | $98.66_{0.23}$ | $99.94_{0.12}$ | $99.94_{0.05}$ | $99.98_{0.04}$ |
| 3 | $99.88_{0.04}$ | $98.44_{0.29}$ | $94.12_{0.77}$ | $85.96_{0.86}$ | $95.86_{0.51}$ | $99.48_{0.07}$ | $99.80_{0.14}$ | $99.88_{0.04}$ |

since lossy embeddings expose detectable patterns that enable targeted attacks. By contrast, lossless schemes show clear superiority: our method improves accuracy by more than 10%. Compared with PRC Watermark, our approach achieves an additional gain of nearly 10% in TPR under Post-Processing distortions, consistent with the main paper. Overall, these findings confirm that *Spherical Watermark* enables exact recovery while maintaining superior robustness over both lossy and lossless baselines.

Table 16: Ablation of parameters $s$ and $N$ on TPR under different attacks. Dataset: SDP. Model: SD v1.5. Case 1: Gaussian Blur, kernel size = 9. Case 2: JPEG-70. Case 3: Brightness, factor = 2.

| Case | sparsity parameter $s$ | | | | repetition count $N$ | | | |
|---|---|---|---|---|---|---|---|---|
| | 1 | 2 | 3 | 4 | 1 | 11 | 21 | 31 |
| 1 | $99.96_{0.05}$ | $99.60_{0.20}$ | $98.06_{0.41}$ | $95.80_{0.97}$ | $98.78_{0.26}$ | $99.92_{0.07}$ | $99.86_{0.05}$ | $99.96_{0.05}$ |
| 2 | $99.86_{0.10}$ | $98.92_{0.27}$ | $96.26_{0.49}$ | $92.62_{0.86}$ | $97.36_{0.46}$ | $99.52_{0.23}$ | $99.94_{0.05}$ | $99.86_{0.10}$ |
| 3 | $99.90_{0.06}$ | $99.04_{0.23}$ | $95.84_{0.62}$ | $91.74_{0.48}$ | $97.24_{0.28}$ | $99.60_{0.13}$ | $99.72_{0.15}$ | $99.90_{0.06}$ |

Table 17: Ablation of parameters $s$ and $N$ on TPR under different attacks. Dataset: SDP. Model: SD v2.1. Case 1: Gaussian Blur, kernel size = 9. Case 2: JPEG-70. Case 3: Brightness, factor = 2.

| Case | sparsity parameter $s$ | | | | repetition count $N$ | | | |
|---|---|---|---|---|---|---|---|---|
| | 1 | 2 | 3 | 4 | 1 | 11 | 21 | 31 |
| 1 | $100.00_{0.00}$ | $99.70_{0.13}$ | $98.60_{0.14}$ | $96.12_{0.48}$ | $99.14_{0.20}$ | $99.94_{0.05}$ | $100.00_{0.00}$ | $100.00_{0.00}$ |
| 2 | $99.96_{0.05}$ | $98.56_{0.19}$ | $94.08_{0.68}$ | $88.44_{0.72}$ | $96.28_{0.50}$ | $99.66_{0.16}$ | $99.82_{0.13}$ | $99.96_{0.05}$ |
| 3 | $99.98_{0.04}$ | $99.20_{0.18}$ | $95.96_{0.39}$ | $89.74_{1.30}$ | $97.22_{0.37}$ | $99.76_{0.16}$ | $99.88_{0.12}$ | $99.98_{0.04}$ |

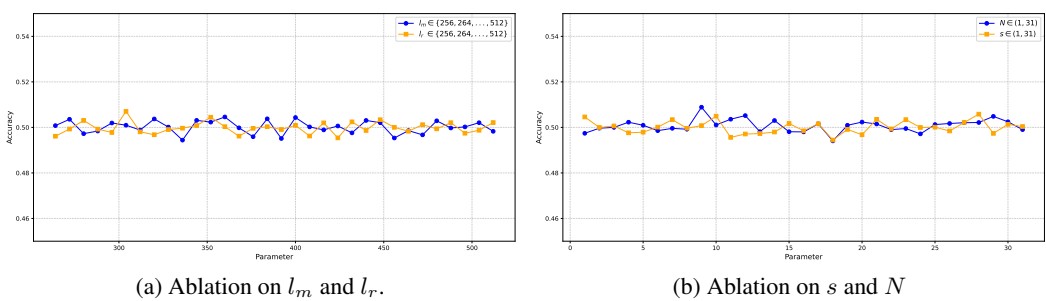

(a) Ablation on $l_m$ and $l_r$.        (b) Ablation on $s$ and $N$

Figure 14: Ablation on hyperparameters undetectability.

## F.5 ABLATION EXPERIMENTS

**Ablation on Modules.** We have supplemented additional module ablation experiments, isolating each module for testing across two datasets and two models. Figure 15 details the experimental results. Here we apply four representative types of attacks: Brightness Adjustment, Gaussian Blur, Median Filter, and Resize. The results clearly show that the absence of the spherical mapping module leads to a significant drop in robustness against all tested attacks. This observation is consistent with the findings discussed in the main paper and underscores the importance of spherical mapping in preserving watermark integrity under perturbation. Furthermore, in the Appendix D, we provide a rigorous analysis of how orthogonal rotation enhances robustness.

**Settings for the Ablation on Parameters.** In the ablation studies, we configure parameters with different settings. To evaluate their impact on undetectability, we set $s = 1, N = 31, l_m = 512$ and $l_r = 512$ in default. Then, we systematically vary a single parameter at a time while keeping the others fixed to the default configuration, allowing us to isolate the impact of each parameter on performance. Note that the size $l_x$ of the latent space varies with parameter adjustments. In Figure 6(d) of the main paper, the values of $s$ and $N$ range from 1 to 31, resulting in 31 test points. For $l_m$ and $l_r$, the values range from 256 to 512 with a step size of 8, yielding 33 test points.

To evaluate the impact of the ablated parameters $s, N, l_m$ on tracing accuracy, we vary one parameter at a time while keeping the latent space size $l_x$ fixed to match the input dimension of diffusion models. Specifically, varying $s$ does not affect other parameters, whereas changing $N$ requires adjusting $l_r$ accordingly, and modifying $l_m$ necessitates updates to both $N$ and $l_r$, so that the constraint $l_x = N \times l_m + l_r$ remains satisfied. This ensures fair comparisons across different ablation settings.

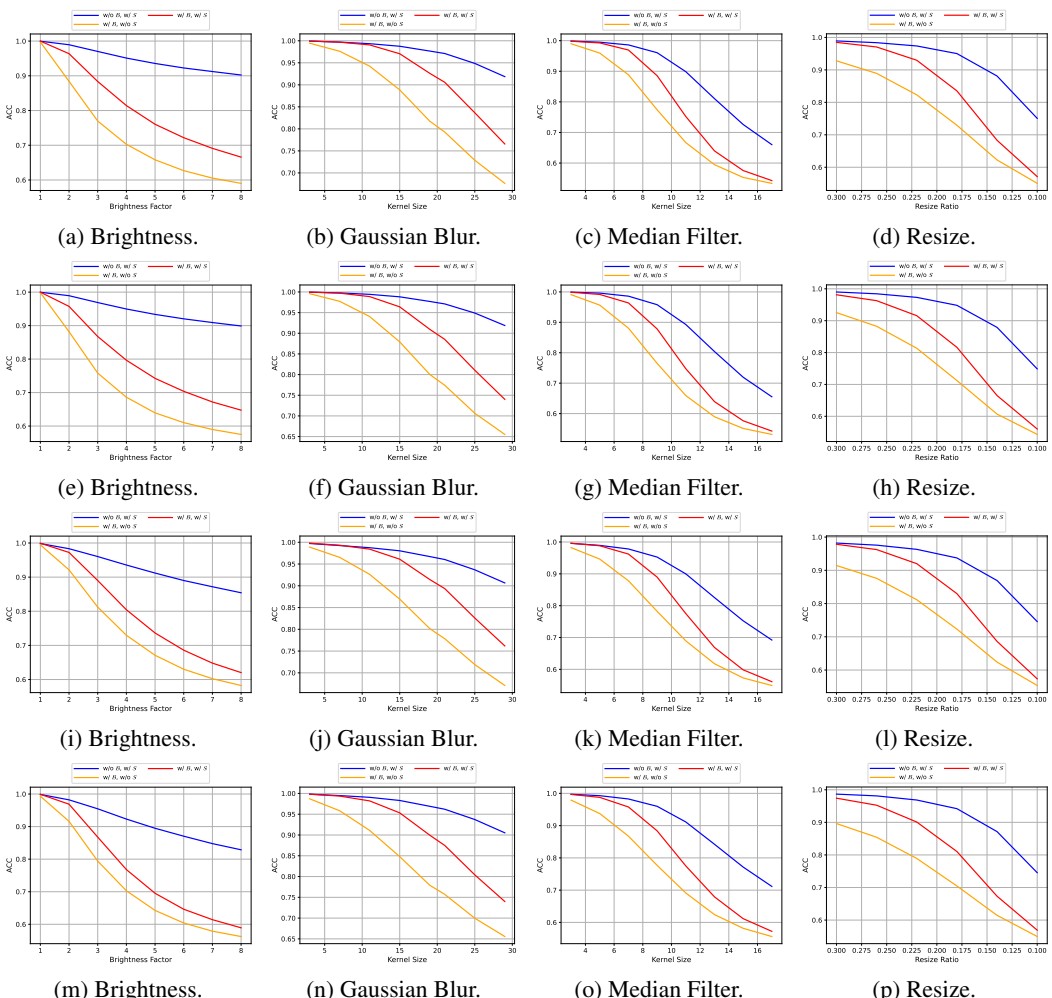

Figure 15: The ACC result of ablation on module $\mathcal{B}$ and $\mathcal{S}$. (a-d) COCO dataset with SD v1.5. (e-h) COCO dataset with SD v2.1. (i-l) SDP dataset with SD v1.5. (m-p) SDP dataset with SD v2.1.

We do not ablate $l_r$ separately, since changes in $l_r$ do not affect the number of watermark–padding mixtures (which is controlled by $s$), and thus have negligible influence on tracing accuracy.

**Ablation on Parameters.** We conduct ablation studies on four hyperparameters: the watermark length $l_m$, padding length $l_r$, row sparsity parameter $s$, and repetition count $N$. To evaluate the undetectability of the four hyperparameters, we conduct experiments in the latent space, as shown in Figure 14. It can be observed that the classification accuracy fluctuates around 50% across all settings, indicating that variations in these parameters do not affect undetectability. In Tables 15, 16, and 17 we report the TPR under various attack scenarios for different values of $s$ and $N$. Consistent with the main paper, increasing $s$ and decreasing $N$ tend to degrade the model's robustness. In Figure 16, we evaluate the impact of watermark length $l_m$, i.e., capacity, across different datasets and models. The attacks include Brightness Adjustment with a factor of 2, Gaussian Blur with kernel size 9, JPEG compression at quality 70, and 30% Resize. The performance of the PRC Watermark drops sharply when the watermark length $l_m$ approaches 2000, whereas the *Spherical Watermark* maintains relatively high ACC and TPR. This demonstrates the design flexibility of our approach, making it well-suited for high-capacity scenarios.

**Settings for the Ablation on Diffusion Sampling Configurations.** In the ablation of diffusion sampling settings studies, we conduct experiment on the COCO dataset using the SD v2.1 model. Here, we evaluate the watermark extraction accuracy under various attacks across three ODE solvers.

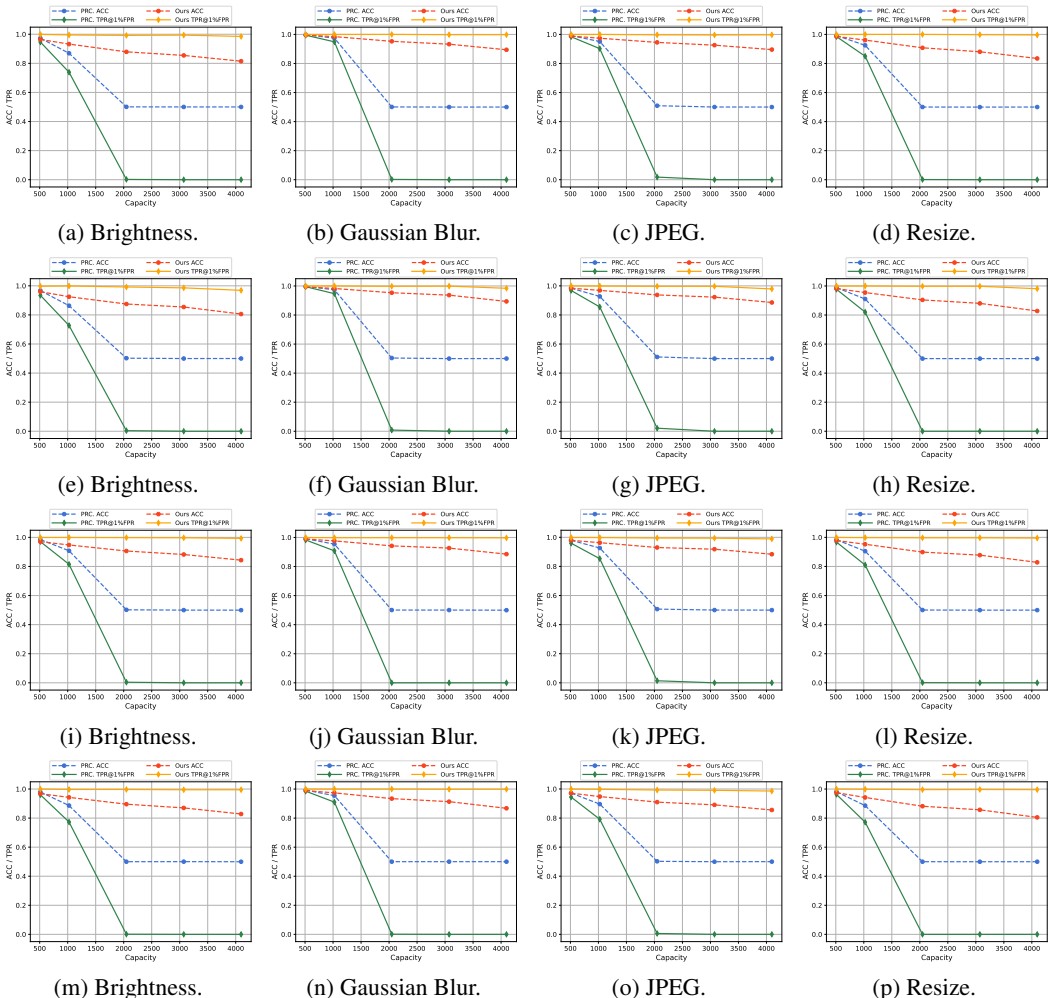

(a) Brightness.  (b) Gaussian Blur.  (c) JPEG.  (d) Resize.

(e) Brightness.  (f) Gaussian Blur.  (g) JPEG.  (h) Resize.

(i) Brightness.  (j) Gaussian Blur.  (k) JPEG.  (l) Resize.

(m) Brightness.  (n) Gaussian Blur.  (o) JPEG.  (p) Resize.

Figure 16: The ACC and TPR of PRC Watermark and Ours under different watermark length $l_m$. (a-d) COCO dataset with SD v1.5. (e-h) COCO dataset with SD v2.1. (i-l) SDP dataset with SD v1.5. (m-p) SDP dataset with SD v2.1.

These attacks include Brightness Adjustment with a factor of 2, Gaussian Blur with a kernel size of 9, JPEG compression with a quality factor of 70 (QF=70), Median Filter with a kernel size of 5, and 50% Resize.

**Ablation on Diffusion Sampling Settings.** To quantitatively assess the impact of inversion errors on watermark extraction, we conduct the ablation study of latent-space perturbation on extraction accuracy. Specifically, we inject Gaussian noise with zero mean and standard deviation $\sigma$ into the watermarked latent vectors, where $\sigma$ controls the noise level. As shown in Figure 17, even at a noise level $1.5\times$ larger than the latent's original standard deviation, the extraction success rate remains above 95%. This confirms that our method is highly tolerant to moderate latent perturbations.

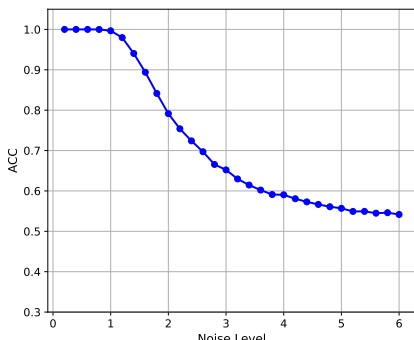

Figure 17: Extraction accuracy for different noise level.

