# OpenReview forum: "Spherical Watermark: Encryption-Free, Lossless Watermarking for Diffusion Models"
_ICLR.cc/2026/Conference — ICLR 2026 Oral_

### Official Review · Reviewer_Hgmm · 2025-10-26

**Soundness:** 3
**Presentation:** 2
**Contribution:** 3
**Rating:** 8
**Confidence:** 2

**Summary:**

This paper introduces Spherical Watermark, a novel encryption-free and lossless watermarking framework for diffusion models designed to overcome the drawbacks of existing methods that either degrade image quality or rely on computationally expensive cryptography. The core contribution is an elegant method that embeds a binary watermark into the model's initial Gaussian noise latent vector. This is achieved first through a binary embedding module that creates a high-entropy bitstream, which is then processed by a spherical mapping module that projects it onto a unit sphere, applies an orthogonal rotation, and scales it with a chi-square distributed radius. This process yields a noise vector that is statistically indistinguishable from a standard Gaussian distribution, a claim supported by theoretical proofs. The framework's key contributions include this new mapping technique, the elimination of cryptographic overhead, and state-of-the-art performance. Experiments show that Spherical Watermark preserves high visual fidelity while offering superior robustness against attacks and a dramatic improvement in computational efficiency, with watermark extraction being orders of magnitude faster than its closest lossless competitor.

**Strengths:**

1.	The core methodology is novel and elegant, using spherical geometry to transform a binary watermark into statistically standard Gaussian noise, which successfully bypasses the need for complex and computationally heavy cryptographic components used in prior lossless methods.
2.	The paper is supported by a strong theoretical foundation, providing formal proofs that the watermarked noise distribution matches a true Gaussian prior up to the third-order moments by leveraging concepts like spherical 3-designs.
3.	The method demonstrates exceptional performance and efficiency, as it is extremely fast in the extraction phase (approximately four orders of magnitude faster than its closest competitor), shows superior robustness against various attacks, and maintains high fidelity and undetectability.
4.	It offers excellent scalability and capacity, handling large watermark payloads without the performance degradation seen in competing approaches, which makes it highly flexible for applications requiring the embedding of rich metadata.

**Weaknesses:**

1.	The comparison of computational efficiency should be included in the main paper instead of the appendix as it can effectively demonstrate the advantages of this method compared to PRC watermark.
2.	Lack of experimental comparison on newer models such as FLUX and Qwen image.
3.	Does this method rely on the accuracy of inversion? I want to know if different inversion methods will affect the accuracy of extraction, and if the sampling step size will also have an impact. In other words, when will the deviation between the latent obtained from inversion and the latent obtained from the original embedding render this method ineffective?

**Questions:**

See the weakness.

---

> ### Author Response · Authors · 2025-11-21
> **Response to Reviewer Hgmm**
>
> Dear Reviewer Hgmm,
>
> Thank you for your thoughtful and detailed review. Your comments are very helpful in improving our work. We hope our responses resolve your concerns.
>
> > Weakness #1: Computational Efficiency Comparison Placement
>
> We thank the reviewer for this helpful suggestion. We have moved the **computational efficiency comparison** to the main text.
>
> > Weakness #2: Results on Newer Models
>
> We thank the reviewer for this comment. We have added experiments on two models, i.e., SD v3 and FLUX.1-DEV. Results are as follows,
>
> Table 1.  Extraction performance on newer models.
>
> | Method     | Metrics | PNG             | Brightness     | Gaussian Blur   | Median Filter   | JPEG           | Resize          |
> | ---------- | ------- | --------------- | -------------- | --------------- | --------------- | -------------- | --------------- |
> | FLUX.1-DEV | ACC     | $99.99_{0.01}$  | $98.84_{0.28}$ | $95.42_{0.28}$  | $91.92_{0.35}$  | $88.85_{0.56}$ | $99.10_{0.13}$  |
> | FLUX.1-DEV | TPR     | $100.00_{0.00}$ | $99.80_{0.40}$ | $100.00_{0.00}$ | $100.00_{0.00}$ | $99.90_{0.20}$ | $100.00_{0.00}$ |
> | SD v3      | ACC     | $99.99_{0.01}$  | $98.67_{0.11}$ | $99.83_{0.10}$  | $99.74_{0.11}$  | $98.97_{0.10}$ | $99.92_{0.06}$  |
> | SD v3      | TPC     | $100.00_{0.00}$ | $99.33_{0.24}$ | $100.00_{0.00}$ | $100.00_{0.00}$ | $99.50_{0.00}$ | $100.00_{0.00}$ |
>
> Specifically, we evaluate several post-processing distortions, including PNG storage, brightness adjustment (×2), Gaussian blur (kernel 9), JPEG (QF=70), median filtering (5×5), and 50% resize. Our method maintains high extraction accuracy on both SD v3 and FLUX.1-DEV, showing strong generalization to newer models. The slightly lower accuracy on FLUX under attacks like JPEG is mainly due to its unstable inversion process, a phenomenon also reported in RF-Solver [1].
>
>
> > Weakness #3: Ablation on Accuracy of Inversion
>
> We thank the reviewer for this insightful question. We have added ablation experiments on different inversion methods, sampling timesteps, and inversion errors.
>
> 1. **Effect of different inversion solvers.**  In Table 2 below, we evaluate the extraction accuracy under three ODE solvers: DDIM, PNDM, and DPM-Solver++. Across all attack types same as Table 1 above, the extraction accuracy remains stable, showing that switching the inversion solver does not introduce meaningful degradation.
>
> Table 2. Ablation results of extraction accuracy on ODE solvers.
>
> | Solver       | PNG            | Brightness     | Gaussian Blur  | Median Filter  | JPEG           | Resize         |
> | ------------ | -------------- | -------------- | -------------- | -------------- | -------------- | -------------- |
> | DDIM         | $99.98_{0.01}$ | $96.06_{0.23}$ | $99.43_{0.02}$ | $99.20_{0.03}$ | $98.39_{0.16}$ | $99.85_{0.01}$ |
> | PNDM         | $99.98_{0.01}$ | $96.17_{0.23}$ | $99.40_{0.02}$ | $99.15_{0.03}$ | $98.41_{0.15}$ | $99.84_{0.01}$ |
> | DPM-Solver++ | $99.98_{0.01}$ | $96.02_{0.26}$ | $99.44_{0.01}$ | $99.21_{0.03}$ | $98.40_{0.15}$ | $99.85_{0.01}$ |
>
> 2. **Effect of generation and inversion timesteps.**  Table 3 below evaluates the impact of varying the number of sampling timesteps for both generation and inversion. The accuracy remains consistently high from 10 to 50 timesteps, indicating that changes in the timestep schedule do not affect watermark recovery.
>
> Table 3.  Ablation results of extraction accuracy on sampling timesteps.
>
> | Gen.\Inv. | 10             | 20             | 30             | 40             | 50             |
> | --------- | -------------- | -------------- | -------------- | -------------- | -------------- |
> | 10        | $99.85_{0.88}$ | $99.92_{0.71}$ | $99.95_{0.57}$ | $99.95_{0.58}$ | $99.95_{0.60}$ |
> | 20        | $99.96_{0.52}$ | $99.97_{0.48}$ | $99.98_{0.35}$ | $99.98_{0.36}$ | $99.98_{0.38}$ |
> | 30        | $99.97_{0.28}$ | $99.99_{0.18}$ | $99.99_{0.16}$ | $99.99_{0.12}$ | $99.99_{0.12}$ |
> | 40        | $99.97_{0.56}$ | $99.98_{0.52}$ | $99.98_{0.48}$ | $99.98_{0.47}$ | $99.98_{0.48}$ |
> | 50        | $99.97_{0.36}$ | $99.98_{0.36}$ | $99.98_{0.29}$ | $99.99_{0.29}$ | $99.99_{0.26}$ |
>
> 3. **Tolerance to inversion errors in latent space.**  To directly quantify how much inversion error our method can tolerate, we inject Gaussian noise into the watermarked latents. As shown in Figure 17, even when the perturbation amplitude reaches 1.5$\times$ the intrinsic standard deviation, the extraction accuracy stays above 95\%. This confirms that our scheme is robust to moderate deviations between the inverted latent and the original embedded latent.
>
> Overall, these experiments show that while our method requires an inversion process, it remains highly tolerant to solver choice, timestep variation, and latent perturbations. Only extremely large distortions, which make diffusion inversion fail, would noticeably affect extraction.
>
> References:
>
> [1] Wang, Pu, et al. "Taming rectified flow for inversion and editing." arXiv:2411.04746.

---

### Official Review · Reviewer_K12q · 2025-10-27

**Soundness:** 3
**Presentation:** 3
**Contribution:** 3
**Rating:** 8
**Confidence:** 2

**Summary:**

This paper introduces Spherical Watermark, an encryption-free and lossless watermarking approach designed for diffusion models. It focuses on tracing and verifying the provenance and authenticity of AI-generated images, addressing recognized limitations of current watermarking techniques, such as quality degradation, detectable shifts, and key management complexity. The method embeds watermarks into the latent noise with indistinguishable Gaussian statistics, utilizing a high-entropy binary embedding and spherical mapping mechanism. The framework maintains perfect image fidelity and allows rapid watermark extraction without modifying the diffusion model. Empirical results demonstrate strong robustness against common image manipulations and adversarial attacks, competitive and often superior to prior approaches. The paper also emphasizes ethical considerations and provides theoretical analysis for transparency and reproducibility.

**Strengths:**

- The proposed watermarking technique preserves image quality, making watermarked and non-watermarked images visually indistinguishable. The framework is efficient, enabling fast watermark extraction with no need for per-image keys or model modifications.
- Robustness is demonstrated against a wide range of image processing operations and adversarial settings.
- The solution is well-supported with both theoretical analysis and comprehensive experiments.
- The method is deployable on mainstream diffusion architectures and easy to integrate in practice. The authors address the ethical implications of watermarking and ensure reproducibility standards.

**Weaknesses:**

- The method may have limitations when facing extremely sophisticated adversarial attacks specifically designed to break watermark recovery.
- There is limited discussion on extending the approach to content editing or direct forgeries, such as partial GAN-based manipulations.
- Some implementation parameters may require careful adjustment for different generative scenarios and applications.

**Questions:**

This is an interesting paper. So I want to know what the main motivation is for the introduction of the spherical mapping module in the context of watermark embedding. Secondly, if this method is so promising, are there any limitations for this method to scale up in practice?

---

> ### Author Response · Authors · 2025-11-21
> **Response to Reviewer K12q (Part 1/2)**
>
> Dear Reviewer K12q,
>
> Thank you for your thoughtful and detailed review. Your comments are very helpful in improving our work. We address each of your points in detail below, and we hope our responses resolve your concerns. Please feel free to ask for clarification or further discussion on any aspect.
>
> > Weakness #1: Robustness against Adversarial Attacks
>
> We thank the reviewer for this comment. Our method is designed to handle realistic distortions encountered in practical deployment, including post-processing and adversarial perturbations. We also test our robustness against re-generation and editing attacks in the revised version, with details provided in Appendix F.2. These experimental results show that our scheme remains robust across these widely adopted attack settings.
>
> The remark on **extremely sophisticated adversarial attacks** corresponds to hypothetical manipulations that aim to break the inversion process. Constructing such targeted attacks would require access to the model’s architecture and parameters, which is unrealistic in practical deployment. In contrast, blind attacks that do not rely on model internals would need to severely distort the image in order to disrupt the diffusion trajectory, and such perturbations typically cause substantial visual or semantic degradation, rendering watermarked images unusable.
>
> To further quantify the robustness of our embedding, in Figure 17 of Appendix F.5, we include an analysis of latent-space deviations. Even when injecting Gaussian noise with amplitude 1.5$\times$ the latent’s intrinsic standard deviation, the extraction accuracy remains above 95\%. This demonstrates that our spherical mapping is highly tolerant to moderate inversion inaccuracies.
>
>
>
> >Weakness #2: Scope of Robustness
>
> We appreciate the reviewer’s insightful comment. In the revised version, we have expanded our robustness evaluation to explicitly include both content editing and direct forgery attacks. As shown in Appendix F.2, our experiments cover a range of diffusion-based and GAN-based manipulations, including MagicBrush, UltraEdit, InstructPix2Pix, and GAN-edit (with representative examples provided in Figure 10). These settings directly target semantic alteration or localized forgery. We present the key results below,
>
> Table 1. Extraction accuracy for image editing attacks.
>
> | Method      | $l_m$ | MagicBrush     | UltraEdit      | InstructPix2Pix | GAN-edit        |
> | ----------- | ----- | -------------- | -------------- | --------------- | --------------- |
> | DwtDct      | 32    | $50.25_{1.28}$ | $49.28_{1.96}$ | $50.56_{1.38}$  | $49.88_{2.21}$  |
> | DwtDctSvd   | 32    | $49.22_{1.76}$ | $49.62_{1.62}$ | $48.53_{0.88}$  | $48.28_{3.44}$  |
> | RivaGan     | 32    | $67.31_{1.24}$ | $52.06_{2.33}$ | $60.56_{1.52}$  | $99.84_{0.14}$  |
> | TrustMark   | 100   | $86.74_{1.85}$ | $68.16_{1.18}$ | $81.68_{3.54}$  | $95.85_{1.34}$  |
> | Robust-Wide | 64    | $94.36_{1.54}$ | $80.09_{0.68}$ | $96.44_{1.10}$  | $98.33_{0.61}$  |
> | PRC         | 100   | $94.14_{2.33}$ | $81.53_{3.09}$ | $83.53_{7.82}$  | $100.00_{0.00}$ |
> | Ours        | 100   | $93.96_{2.94}$ | $86.11_{1.61}$ | $92.53_{2.38}$  | $100.00_{0.00}$ |
>
> Across all editing and forgery manipulations, our method maintains extraction accuracy above 85\% and performs comparably to recent robust watermarking methods such as TrustMark [1] and Robust-Wide [2]. This indicates that our approach generalizes well to content-altering operations rather than only to post-processing or adversarial perturbations.
>
>
>
> > Weakness #3: Implementation Parameters Adjustment
>
> We thank the reviewer for raising this concern. The implementation parameters in our method are determined in a simple and systematic manner:
>
> 1. **Fix the latent dimension $l_x$** once the generative model is chosen.
> 2. **Choose the desired watermark length $l_m$**, which specifies how many bits to embed.
> 3. **Set the sparsity parameter $s = 1$**, a stable choice that we use for all experiments.
> 4. **Determine the repetition factor $N$** by computing ${N = \lfloor l_x / l_m \rfloor}$ and reducing it slightly if necessary to ensure a nontrivial padding space  ${l_r = l_x - N \times l_m \geq N \times s}$.
> 5. **Select the rotation dimension $l_C$**. It only needs to be an integer divisor of $l_x$, which allows the rotation to be applied block-wise.
>
> All remaining diffusion-related parameters (e.g., ODE solver, guidance scale) use the model’s native configuration and require no additional tuning. Overall, the parameter setup is straightforward and does not introduce application-specific complexity.
>
> References:
>
> [1] Bui, Agarwal, et al. "TrustMark: Universal Watermarking for Arbitrary Resolution Images." ICCV 2025.
>
> [2] Hu, Zhang, et al. "Robust-wide: Robust watermarking against instruction-driven image editing." ECCV 2024.

---

> > ### Author Response · Authors · 2025-11-21
> > **Response to Reviewer K12q (Part 2/2)**
> >
> > > Question #1: Motivation and Scalability
> >
> > We thank the reviewer for the positive feedback and for raising these two insightful questions.
> >
> > 1. **Motivation of spherical mapping.**  Our goal is to ensure that the watermarked noise preserves the standard Gaussian distribution. We begin by observing, via **Lemma 3.4**, that a multivariate standard Gaussian vector can be decomposed into a uniformly distributed direction on the sphere and an independent chi–square norm. This motivates performing watermark embedding on the **spherical component**. To obtain an approximately uniform spherical distribution, we leverage the theory of **spherical $t$-designs** and construct our mapping accordingly. Through theoretical analysis, we further show that the mapped distribution matches the standard Gaussian distribution up to third-order moments, making the watermarked noise statistically indistinguishable from true Gaussian samples. This guarantees distributional consistency and naturally enhances the robustness of our watermarking scheme (See Appendix D for detailed analysis).
> >
> > 2. **Scalability and practical limitations.**  In practice, our method can be generalized to many generative models. In Appendix F.1, the Spherical Watermark applies seamlessly to newer transformer-based latent diffusion architectures such as SD v3 and FLUX.1-DEV, as well as to pixel-space diffusion models and flow-based models. Across all these settings, we observe consistently reliable embedding and extraction performance. The main practical constraint lies in the requirement for the accurate inversion procedure. Models that do not support inversion cannot directly incorporate our mechanism. More generally, our approach applies to any generative model that satisfies two conditions:
> >    (1) sampling from a Gaussian prior, and
> >    (2) providing an invertible mapping between the image and noise domains.
> >    These conditions hold for a broad family of architectures, including diffusion models, normalizing flows, and GAN frameworks equipped with effective inversion methods.

---

> > > ### Comment · Reviewer_K12q · 2025-11-26
> > > **Thanks for addressing the questions.**
> > >
> > > Thanks to the authors for addressing my concerns and questions in detail! I would like to keep my original rating.

---

### Official Review · Reviewer_vTJw · 2025-10-29

**Soundness:** 4
**Presentation:** 3
**Contribution:** 3
**Rating:** 8
**Confidence:** 3

**Summary:**

This paper proposes a training-free and lossless diffusion watermarking method that encodes a binary watermark message into the initial noise of a diffusion model. The method consists of two main transformation steps. First, the message is encrypted by mixing it with random padding through a structured binary embedding matrix. Second, the resulting vector is projected onto a unit sphere and then transformed by a random orthogonal rotation followed by a radius rescaling using a chi distribution. This produces a final watermark vector that closely matches the distribution of standard Gaussian noise, making it suitable as the initial noise for diffusion models and statistically indistinguishable from normal noise. The method is compatible with multiple diffusion models, does not modify model parameters, and enables fast decoding.

**Strengths:**

1. The paper is well motivated and logically structured. The paper addresses the problem of embedding watermarks in diffusion-generated images in a clean and motivated way.
2. The proposed method is reasonable and effective. The approach avoids any model fine-tuning or training and leverages statistical geometry (spherical design + chi rescaling) to achieve high-quality watermarking while maintaining indistinguishability.
3. The experiments are comprehensive. The paper evaluates watermark accuracy and detectability under different attacks. The ablation study also demonstrates the effectiveness of their key designs.
4. The paper is easy to read and well-organized, with clear diagrams and concrete definitions.

**Weaknesses:**

1. Lack of diffusion-based attacks: While the paper evaluates robustness under post-processing and adversarial attacks, it does not include experiments on regeneration or rinse-based attacks (e.g., re-diffusion or editing using other diffusion models), which have been recently identified as strong attacks for watermark removal. I suggest the authors refer to arXiv:2401.08573 and consider incorporating some of their benchmarking strategies.

2. Storage overhead for decoding: To decode the watermark, the user must store the embedding matrix $T$ (specifically, the sparse matrix
$R$) and the rotation matrix $C$. Since $R \in \mathbb{F}_2 ^{N l_m \times l_r}$ and $C \in \mathbb{R}_2 ^{l_x \times l_x}$, the memory cost could be significant, especially when generating high-resolution images with large latent dimensions. A naïve implementation would incur nontrivial storage. Discussion about the storage overhead should be included in the main text.

**Questions:**

1. Have you evaluated your method under regeneration-based or rinse attacks? If not, can you comment on the potential vulnerability under such transformations?

2. Could you elaborate on whether $T$ and $C$ are reused across images, or whether they are derived per image? What is the typical memory overhead for storing or generating them in a realistic deployment?

---

> ### Author Response · Authors · 2025-11-21
> **Response to Reviewer vTJw**
>
> Dear Reviewer vTJw,
>
> Thank you for your thoughtful and detailed review. Your comments are very helpful in improving our work. We address each of your points in detail below, and we hope our responses resolve your concerns. Please feel free to ask for clarification or further discussion on any aspect.
>
> >Weakness #1 & Question #1: Scope of Robustness
>
> We appreciate the reviewer’s insightful comment. In response, we have added comprehensive evaluations on re-generation and rinse attacks in Appendix F.2 of the revised version. Our re-generation setup follows the WAVES benchmark (arXiv:2401.08573), including Regen-Diff, Rinse-2xDiff, and Regen-VAE. In addition, we also include experiments on robustness against image editing attacks. We evaluate under MagicBrush, UltraEdit, and InstructPix2Pix from W-Bench [1] benchmark. The key quantitative results are summarized below.
>
> Table 1. Extraction accuracy for re-generation and editing attacks.
>
> | Method      | $l_m$ | Regen-Diff     | Rinse-2xDiff   | Regen-VAE      | MagicBrush     | UltraEdit      | InstructPix2Pix |
> | ----------- | ----- | -------------- | -------------- | -------------- | -------------- | -------------- | --------------- |
> | DwtDct      | 32    | $49.96_{0.73}$ | $49.91_{0.46}$ | $50.27_{0.72}$ | $50.25_{1.28}$ | $49.28_{1.96}$ | $50.56_{1.38}$  |
> | DwtDctSvd   | 32    | $50.20_{0.93}$ | $49.69_{0.74}$ | $48.98_{0.58}$ | $49.22_{1.76}$ | $49.62_{1.62}$ | $48.53_{0.88}$  |
> | RivaGan     | 32    | $56.77_{0.56}$ | $54.19_{0.53}$ | $50.89_{0.33}$ | $67.31_{1.24}$ | $52.06_{2.33}$ | $60.56_{1.52}$  |
> | TrustMark   | 100   | $71.36_{0.46}$ | $59.94_{0.30}$ | $93.87_{0.33}$ | $86.74_{1.85}$ | $68.16_{1.18}$ | $81.68_{3.54}$  |
> | Robust-Wide | 64    | $96.90_{0.18}$ | $93.22_{0.23}$ | $94.15_{0.63}$ | $94.36_{1.54}$ | $80.09_{0.68}$ | $96.44_{1.10}$  |
> | PRC         | 100   | $99.26_{0.41}$ | $94.22_{0.88}$ | $75.91_{1.25}$ | $94.14_{2.33}$ | $81.53_{3.09}$ | $83.53_{7.82}$  |
> | Ours        | 100   | $99.63_{0.17}$ | $97.70_{0.37}$ | $87.48_{0.45}$ | $93.96_{2.94}$ | $86.11_{1.61}$ | $92.53_{2.38}$  |
>
> 1. **Re-generation**. As shown in Figure 9 of the appendix, re-generation attacks preserve the overall image content while introducing varying degrees of reconstruction noise. Our method achieves the highest accuracy under both Regen-Diff and Rinse-2xDiff, delivering superior robustness to diffusion-based re-generation attacks. In the Regen-VAE case, TrustMark [2] and Robust-Wide [3] obtain slightly higher accuracy, while our method still outperforms the lossless PRC Watermark across all attack settings.
>
> 2. **Editing**. As illustrated by the edited examples in Figure 10 of the appendix, editing or forgery operations may substantially alter the image content. Across these manipulations, our method performs comparably to recent robust watermarking approaches and maintains accuracy above 85% in all cases.
>
> These results indicate that our robustness against image re-generation and editing, is broadly comparable to existing state-of-the-art watermarking methods.
>
>
>
> > Weakness #2 & Question #2: Storage Overhead of  T and C
>
> We thank the reviewer for raising this point. In our framework, both the embedding matrix **T** and the rotation matrix **C** are generated once during the model build phase and are **reused across all images**. The rotation matrix **C** is deterministically reconstructed from a random seed using Gaussian initialization and QR decomposition, while **T** only requires storing the index set **P** returned by *Algorithm 1*. As shown in our supplementary materials, the entire **signature file** (containing both the seed and index set) occupies only **454 KB**, which makes the storage cost practically negligible.
>
> In realistic deployment, no per-image generation or storage is required. The matrices are loaded once and reused throughout inference. Both storage and generation costs are minimal compared to the diffusion model parameters and do not impact deployment efficiency.
>
>
>
> References:
>
> [1] Lu, Zhou, et al. "Robust Watermarking Using Generative Priors Against Image Editing: From Benchmarking to Advances." ICLR 2025.
>
> [2] Bui, Agarwal, et al. "TrustMark: Universal Watermarking for Arbitrary Resolution Images." ICCV 2025.
>
> [3] Hu, Zhang, et al. "Robust-wide: Robust watermarking against instruction-driven image editing." ECCV 2024.

---

> ### Comment · Reviewer_vTJw · 2025-11-27
>
> Thank you for the clarification about the robustness to regeneration-based attacks and key storage overhead. The authors have addressed my questions. I would like to maintain my original positive assessment and score.

---

### Official Review · Reviewer_YSkV · 2025-10-31

**Soundness:** 3
**Presentation:** 3
**Contribution:** 4
**Rating:** 6
**Confidence:** 3

**Summary:**

This paper addresses the critical need for provenance in diffusion-generated images by proposing a new watermarking scheme. The authors introduce "Spherical Watermark," a lossless and encryption-free framework that embeds a binary watermark into the initial Gaussian noise by mixing it with random padding, projecting it onto a unit sphere, applying an orthogonal rotation, and scaling it with a chi-square-distributed radius. The method is theoretically proven and empirically demonstrated to be statistically indistinguishable from standard Gaussian noise, while also being computationally efficient and robust to various post-processing and adversarial attacks, outperforming prior lossless methods.

**Strengths:**

S1 (technical novelty): The proposed spherical mapping module is a novel technical contribution. It provides a clear mathematical pipeline to transform a structured binary vector into a vector that is statistically indistinguishable from a standard Gaussian distribution.

S2 (theoretical foundation): The "lossless" claim is strongly supported by a rigorous theoretical analysis. The paper proves that the watermarked noise distribution matches a true Gaussian prior up to third-order moments by leveraging the properties of spherical 3-designs.

S3 (strong performance): The paper shows clear improvements in efficiency and robustness over baseline methods. The method is also shown to be effectively indistinguishable from the original, non-watermarked distribution.

S4 (comprehensive experiments): The ablation studies clearly justify the design, demonstrating the necessity of both the binary embedding and spherical mapping modules for undetectability and robustness, respectively.

**Weaknesses:**

W1 (motivation): The paper claims that existing methods require per-image key storage or cryptographic overhead, but this method also has cryptographic overhead (Eq. 13). Hence, The "encryption-free" claim is potentially misleading.

W2 (scope of robustness): The paper does not explicitly test against attacks that are more specific to generative models, such as watermark destruction via diffusion-inversion and re-generation with different noise. The authors have acknowledged that resisting editing/forgery is a limitation but is out of scope.

W3 (missing related work): Some recent works are not discussed, such as [1,2].

- [1] Wei et al. Robust watermarking for diffusion models: A unified multi-dimensional recipe. 2025.
- [2] Wang et al. SleeperMark: Towards Robust Watermark against Fine-Tuning Text-to-image Diffusion Models. 2025.

**Questions:**

See weaknesses.

---

> ### Author Response · Authors · 2025-11-21
> **Response to Reviewer YSkV**
>
> Dear Reviewer YSkV,
>
> Thank you for your thoughtful and detailed review. Your comments are very helpful in improving our work. We address each of your points in detail below, and we hope our responses resolve your concerns. Please feel free to ask for clarification or further discussion on any aspect.
>
> > Weakness #1: Concerns about Encryption-free
>
> We thank the reviewer for raising this point. Our use of *encryption-free* refers to the absence of **per-image key management, key storage, and cryptographic encoding/decoding**  that prior watermarking schemes rely on. In our framework, the signature {**T**, **C**} is generated once during the system build phase and **reused across all images**. The operations in Eq. (9)–(13) are purely **deterministic invertible transformations** combined with rounding and aggregation, and do **not** involve encryption, decryption, key exchange, or any cipher-based primitives. The *encryption-free* term is therefore intended to indicate that our design eliminates **per-image cryptographic overhead** while preserving verifiable and lossless decoding.
>
>
>
> >Weakness #2: Scope of Robustness
>
> We appreciate the reviewer’s insightful comment. We have expanded our evaluation to include both re-generation and editing attacks in Appendix F.2 of the revised version. For clarity, we also present the key experimental results below.
>
> Table 1. Extraction accuracy for re-generation and editing attacks.
>
> | Method      | $l_m$ | Regen-Diff     | Rinse-2xDiff   | Regen-VAE      | MagicBrush     | UltraEdit      | InstructPix2Pix | GAN-edit        |
> | ----------- | ----- | -------------- | -------------- | -------------- | -------------- | -------------- | --------------- | --------------- |
> | DwtDct      | 32    | $49.96_{0.73}$ | $49.91_{0.46}$ | $50.27_{0.72}$ | $50.25_{1.28}$ | $49.28_{1.96}$ | $50.56_{1.38}$  | $49.88_{2.21}$  |
> | DwtDctSvd   | 32    | $50.20_{0.93}$ | $49.69_{0.74}$ | $48.98_{0.58}$ | $49.22_{1.76}$ | $49.62_{1.62}$ | $48.53_{0.88}$  | $48.28_{3.44}$  |
> | RivaGan     | 32    | $56.77_{0.56}$ | $54.19_{0.53}$ | $50.89_{0.33}$ | $67.31_{1.24}$ | $52.06_{2.33}$ | $60.56_{1.52}$  | $99.84_{0.14}$  |
> | TrustMark   | 100   | $71.36_{0.46}$ | $59.94_{0.30}$ | $93.87_{0.33}$ | $86.74_{1.85}$ | $68.16_{1.18}$ | $81.68_{3.54}$  | $95.85_{1.34}$  |
> | Robust-Wide | 64    | $96.90_{0.18}$ | $93.22_{0.23}$ | $94.15_{0.63}$ | $94.36_{1.54}$ | $80.09_{0.68}$ | $96.44_{1.10}$  | $98.33_{0.61}$  |
> | PRC         | 100   | $99.26_{0.41}$ | $94.22_{0.88}$ | $75.91_{1.25}$ | $94.14_{2.33}$ | $81.53_{3.09}$ | $83.53_{7.82}$  | $100.00_{0.00}$ |
> | Ours        | 100   | $99.63_{0.17}$ | $97.70_{0.37}$ | $87.48_{0.45}$ | $93.96_{2.94}$ | $86.11_{1.61}$ | $92.53_{2.38}$  | $100.00_{0.00}$ |
>
> 1. **For re-generation attacks**, we follow the WAVES benchmark [1], which includes Regen-Diff, Rinse-2xDiff, and Regen-VAE. Representative regenerated samples are shown in Figure 9 of the appendix. Our method achieves the highest accuracy under both Regen-Diff and Rinse-2xDiff, delivering superior robustness to diffusion-based re-generation attacks. In the Regen-VAE case, TrustMark [2] and Robust-Wide [3] obtain slightly higher accuracy, while our method still outperforms the lossless PRC Watermark across all attack settings.
>
> 2. **For image editing and instruction-driven forgery**, we evaluate under MagicBrush, UltraEdit, InstructPix2Pix, and GAN-edit. Example edited images appear in Figure 10 of the appendix. Across these manipulations, our method performs comparably to recent robust watermarking approaches and maintains accuracy above 85% in all cases.
>
> These results indicate that our robustness against image re-generation and editing, is broadly comparable to existing state-of-the-art watermarking methods.
>
>
>
> > Weakness #3: Missing Related Works
>
> We thank the reviewer for pointing out the missing references. In the revised version, we have added discussions of both Wei et al. (2025) and SleeperMark (2025). These additions improve the completeness of our literature review. We appreciate the reviewer for bringing these works to our attention.
>
>
>
> References:
>
> [1] An, Ding, et al. "WAVES: Benchmarking the Robustness of Image Watermarks. " ICML 2024.
>
> [2] Bui, Agarwal, et al. "TrustMark: Universal Watermarking for Arbitrary Resolution Images." ICCV 2025.
>
> [3] Hu, Zhang, et al. "Robust-wide: Robust watermarking against instruction-driven image editing." ECCV 2024.

---

> > ### Comment · Reviewer_YSkV · 2025-11-22
> >
> > Thank you for the detailed response. My concerns have been addressed. I will keep my positive score.

---

### Author Response · Authors · 2025-11-21
**General Responses and Revision Summary**

We sincerely thank all reviewers for their valuable comments and constructive suggestions. We have revised the paper accordingly, with all updates highlighted in **blue** in the revised version. The feedback has greatly helped us improve the clarity, completeness, and experimental evaluation of our work. We summarize the main changes below:

1. **Robustness under re-generation and editing attacks (Appendix F.2).**
   Following the comments from reviewers *YSkV*, *vTJw*, and *K12q*, we added comprehensive experiments on re-generation attacks (Regen-Diff, Rinse-2xDiff, Regen-VAE) and image editing (MagicBrush, UltraEdit, InstructPix2Pix, GAN-edit). These results show that our method remains traceable under these attacks.

2. **Evaluation on newer generative models (Appendix F.1).**
   As suggested by reviewer *Hgmm*, we extended our study to modern latent-space diffusion architectures, including **Stable Diffusion v3** and **FLUX.1-DEV**, demonstrating that our approach generalizes well to transformer-based diffusion models.

3. **Ablation studies on sampling configurations and inversion accuracy (Sec. 4.3 and Appendix F.5).**
   As suggested by reviewer *Hgmm*, we added ablations covering (i) different ODE solvers, (ii) generation/inversion timesteps, and (iii) latent-space perturbations. These results show that the extraction accuracy is stable across solvers, timestep choices, and moderate inversion errors.

4. **Improved writing quality and organization.**
   We refined our paper in terms of clarity and formatting, following reviewer suggestions.

We appreciate the reviewers’ thoughtful feedback and positive recognition. Their suggestions significantly strengthened this paper, and we hope the revisions adequately address all concerns.

---

### Comment · Area_Chair_MT5Y · 2025-11-25

Dear Reviewers,

The authors have submitted their responses to your questions and feedbacks. Please read them and give your comments.

Regards, AC

---

### Author Response · Authors · 2025-12-04
**Summary of Revisions and Reviewer Feedback (Post-Rebuttal)**

Dear AC,

We sincerely appreciate the time and effort you are dedicating to our submission, especially given the additional workload caused by the recent OpenReview system issue. We also thank all reviewers for their valuable comments.



In this paper, we propose **Spherical Watermark**, a lossless and encryption-free latent watermarking framework designed for reliable provenance tracing in diffusion models. Below we provide a brief summary of the **positive aspects** noted by the reviewers:

1. **Novelty.** We introduce a new spherical mapping mechanism that enables model developers to trace misuse and supports high-capacity watermark embedding. The novelty of our method was mentioned by **all reviewers (*YSkV, vTJw, K12q, Hgmm*)**.
2. **Theoretical foundation.** We provide a rigorous mathematical analysis showing that our embedding is lossless and matches the Gaussian prior up to the third moment. The strength of theoretical support was mentioned by **all reviewers (*YSkV, vTJw, K12q, Hgmm*)**.
3. **Experimental evaluation.** We conduct extensive experiments demonstrating strong fidelity, robustness, undetectability, and detailed ablations. **All reviewers** commented that these empirical results provide solid support for the method ***(YSkV, vTJw, K12q, Hgmm)***.
4. **Efficiency and practical applicability.** Our method is training-free, scalable across generative architectures, supports flexible watermark lengths, and enables high efficiency. These practical benefits were noted by **reviewers (*K12q, Hgmm*)**.



We sincerely thank all reviewers for their thoughtful and constructive feedback during the rebuttal phase. Their comments were highly valuable in improving our paper. In response, we revised the manuscript accordingly, with all changes highlighted in **blue** in the revised version. Below, we summarize ***all concerns*** raised by the reviewers and describe the corresponding revisions made in the revised PDF.

> (A). Additional Experiments Added After Rebuttal

1. **Robustness under re-generation and editing attacks (Appendix F.2).**
   Following the comments from **reviewers *YSkV*, *vTJw*, and *K12q*,** we added comprehensive experiments on re-generation attacks (Regen-Diff, Rinse-2xDiff, Regen-VAE) and image editing (MagicBrush, UltraEdit, InstructPix2Pix, GAN-edit). These results show that our method remains traceable under these attacks.
2. **Evaluation on newer generative models (Appendix F.1).**
   As suggested by **reviewer *Hgmm***, we extended our study to modern latent-space diffusion architectures, including **Stable Diffusion v3** and **FLUX.1-DEV**, demonstrating that our approach generalizes well to transformer-based diffusion models.
3. **Ablation studies on sampling configurations and inversion accuracy (Sec. 4.3 and Appendix F.5).**
   As suggested by **reviewer *Hgmm***, we added ablations covering (i) different ODE solvers, (ii) generation/inversion timesteps, and (iii) latent-space perturbations. These results show that the extraction accuracy is stable across solvers, timestep choices, and moderate inversion errors.

> (B). Response to Individual Reviewers

1. **Response to Reviewer *YSkV*.** We clarified the meaning of "encryption-free", added comprehensive re-generation and editing robustness experiments (see (A).1 of this summary), and incorporated the missing related works  into Sec. 2 (L81–L90) of the revised PDF, suggested by the reviewer.
2. **Response to Reviewer *vTJw*.** We expanded the robustness evaluation to include re-generation and editing attacks (see (A).1), and clarified the storage and reuse of the matrices **T** and **C**, showing that the overhead is negligible.
3. **Response to Reviewer *K12q*.** We expanded robustness experiments to include re-generation and editing attacks (see (A).1). We discussed the feasibility of adversarial inversion-breaking attacks, and added ablations on inversion accuracy to support our conclusion regarding inversion robustness (see (A).3), and provided clearer explanations on parameter selection and the motivation of spherical mapping.

4. **Response to Reviewer *Hgmm*.** We moved the computational efficiency comparison into the main paper, added experiments on SD v3 and FLUX.1-DEV (see (A).2), and included ablations on ODE-solver choice, timestep variation, and inversion errors to clarify the method’s error tolerance (see (A).3).


In addition, we want to note that ***Reviewers YSkV, vTJw, and K12q*** had already indicated that their concerns were addressed before **Nov 26, 23:59 EST**, which was earlier than the OpenReview issue notified on **Nov 27, 10:09 EST** by the ICLR team. We sincerely hope that the clarifications and revisions provided during the rebuttal phase adequately address all reviewer concerns. Thank you again for your time and consideration.

Best regards,

The authors of submission14944

---

### Meta-Review · Area_Chair_mDyQ · 2025-12-25

**Summary:**

This paper received very high ratings, 6, 8, 8 and 8. The reviewers appreciated the novelty, the theoretical justification, the scalability and performance of the work. The reviewers commented on the evaluation such as the lack of re-generation, editing and adversarial attacks, and terms. During the rebuttal period, the authors submitted detailed responses to address reviewers’ concerns. Reviewers, K12q, vTJw and YSkV confirmed that the authors addressed their concerns and questions. The AC has read the comments and responses and agrees that the reviewers’ concerns have been well addressed. Thus, the AC recommends accepting this paper.

**Reviewer Concerns:**

Reviewers, K12q, vTJw and YSkV confirmed that the authors addressed their concerns and questions.

**Reviewer Scores:**

The original scores are very high. No reviewer would change score.

---

### Decision · Program_Chairs · 2026-01-26

Accept (Oral)